geochemistry

turquoise, ED-XRF, UV–Vis spectra, FTIR, colour difference

**Author for correspondence:**
Ying Guo
e-mail: guoying@cugb.edu.cn

# The impact of trace metal cations and absorbed water on colour transition of turquoise

## Xueding Wang and Ying Guo

School of Gemmology, China University of Geosciences (Beijing), Beijing 100083, People's Republic of China

(iD) XW, 0000-0003-0273-4289; YG, 0000-0002-7208-4368

Thirty-five gem-quality turquoise samples with various colours were investigated using energy-dispersive X-ray fluorescence spectroscopy, ultraviolet–visible spectroscopy, Fourier-transform infrared spectroscopy and scanning electron microscopy. Sample chemical and spectral analyses indicate that $Fe^{3+}$ contributes to green hue of turquoise, whose absorption band exhibits a bathochromic shift from 426 to 428 nm with increasing V content in the solid-solution series turquoise-chalcosiderite. $V^{3+}$ enhances absorption in the blue and orange regions, and $Cr^{3+}$ increases absorption in the green region, both of which are responsible for the vivid greenish yellow in faustite. Substitutions of Al by medium-sized trivalent cations (primarily $Fe^{3+}$ and $V^{3+}$) enhance polarity of the phosphate group $(PO_4)^{3-}$, resulting in strong absorption in the infrared spectra for analogues of turquoise. The reflectivity ratio ($R_{OH}$) of the double absorption peaks at 781 and 833 $cm^{-1}$ allows faustite to be distinguished from turquoise and chalcosiderite, with a value greater than 1, while V-rich faustite only has a single absorption peak at 798 $cm^{-1}$. An increasing amount of absorbed water contributes to blue chroma in turquoise and has a negative effect on lightness based on the CIE 1976 $L^*a^*b^*$ colour system. Loose turquoise with a low specific gravity tends to display greater colour differences with a significant decrease in lightness.

## 1. Introduction

Turquoise is one of the most well-known and ancient types of jade, prevalent in Chinese literature, religion, politics and arts (e.g. jewellery) [1]. Turquoise deposits are well developed in the United States [2,3], Egypt [4], Iran [5] and China, where turquoise in China is mostly mined in the Hubei and Anhui

This article has been edited by the Royal Society of Chemistry, including the commissioning, peer review process and editorial aspects up to the point of acceptance.

provinces. The mineralization of turquoise commonly occurs in the tectonic fracture zone within thick- and thin-bedded siliceous and carbonaceous-siliceous slates, while the gem-quality turquoise is almost formed by exogenous leaching [6].

Turquoise, a hydrated copper aluminium phosphate, is a main mineral in the turquoise group which consists of at least six isostructural end-members, including faustite (the rare Zn analogue of turquoise) and chalcosiderite (the Fe analogue of turquoise) [7]. A complex chemical composition produces a wide colour range in turquoise [8]. The general formula for the turquoise group can be written as $A_{0-1}B_6(PO_4)_4(OH)_8 \cdot 4H_2O$ with $Cu^{2+}$ or $Fe^{2+}$ as the most popular constituents at the A-position and $Al^{3+}$ and $Fe^{2+}$ at the B-position. However, $Zn^{2+}$ or $Ca^{2+}$ can be present at the A-position and $V^{3+}$ or $Cr^{3+}$ can occur at the B-position in some rare turquoises [9]. Previous research suggests that the chemical composition, structure compactness and adsorbed water content of turquoise are the main internal factors affecting its colour—explained by crystal field theory and charge transferring [10]. Turquoise blue, for example, comprises Cu octahedra in crystalline turquoise, and a colour transition from blue to yellow is determined by Fe content [11]. The substitution of Al by $Fe^{3+}$ results in a yellow hue and an upward trend in Fe content can create brown turquoise [12]. However, there are other trace elements found in turquoise, which may affect the colour of turquoise. Ultraviolet–visible spectroscopy (UV–Vis) probes the absorption behaviours of different metal cations, and is an effective technique in establishing the impact of transition metal cations on a gem's colour. UV–Vis lends itself to the study of colours in various gemstones, such as emerald [13], variscite [14], demantoid [15], uvarovite [16] and turquoise [6,17].

Currently, several techniques can be applied to investigate the chemical composition of turquoise. Secondary ion mass spectrometry (SIMS) and multi-receiver inductively coupled plasma mass spectrometry (MC-ICP-MS) are used for the isotopic analysis of hydrogen, copper, strontium and lead in turquoise, distinguishing the provenance of turquoise artefacts [18]. Laser ablation inductively coupled plasma emission spectrometry (LA-ICP-AES) is used to study the geochemical characteristics of trace elements and rare-earth elements in modern turquoise, to determine its origin [19]. Electron microprobe analysis (EMPA) is used to quantitatively analyse the main elements and rare earth elements in turquoise, especially for the turquoise of various colours [20]. Compared with traditional methods, energy-dispersive X-ray fluorescence (ED-XRF) is an advanced and non-destructive technique, which is widely applied in chemical analyses for mineralogy [21], materials science [22], geology [23], environmental science [24] and gemmology. ED-XRF is similarly suitable for the study of turquoise [25–27]. Several scholars have used LA-ICP-MS to measure the element content of uniformly coloured turquoise, establishing a standard working curve in order to predict the element content of unknown turquoise using ED-XRF [28]. In addition, ED-XRF can be used to quantitatively or semi-quantitatively analyse differences in the element content of turquoise and its imitators, allowing the identification of natural turquoise [29]. Fourier-transform infrared (FTIR) spectroscopy is a valuable technique in studying hydrated minerals, and is sensitive to the hydroxyl (OH) group; this type of spectroscopy easily distinguishes OH with $H_2O$ molecules in the structure. Moreover, FTIR is well used to determine the origin of natural turquoise [30–32].

The type and content of water in turquoise can significantly influence its physical properties, especially its colour. Three types of water are found in turquoise, including hydroxyl groups with strong hydrogen bonds (Al–OH), $[Cu(H_2O)_4]^{2+}$ with relatively weak hydrogen bonds and adsorbed water in pores or micro-fractures, whose content generally influences colour [33]. The content of adsorbed water in natural turquoise is usually less than 2% and varies as a function of environmental humidity [34]. However, it is poorly understood what role an increase in absorbed-water content plays in turquoise colour, and most previous research is based on naked-eye observation, lacking quantitative analysis. Therefore, we make good use of colorimetry theory to design a water immersion experiment.

The CIE 1976 $L^*a^*b^*$ uniform colour system is the most popular system for colour measurement and analysis, currently recommended by the International Commission on Illumination (CIE) [35–38]. Serving as the foundation for the quantitative characterization of colour, this colour system possesses good colour uniformity, where the colour difference observed by the naked eye is positively correlated with that calculated by colorimetric coordinates in this system [39,40]. The colour system is widely applied in the study of gem's colour, including jadeite-jade [41–43], peridot [44,45], tourmaline [46], amethyst [47] and diamond [48]. The formula CIE DE2000 can be used to express colour difference quantitatively and effectively [49–51]. Therefore, this formula was applied to calculate the colour differences in turquoise caused by water immersion, as well as study the impact of absorbed water on turquoise colour.

# 2. Material and methods

## 2.1. Material

We collected 35 pieces of gem-quality turquoise samples from China, ranging from 0.20 to 0.40 ct, displaying a colour transition from blue to greenish yellow. Fifteen samples of turquoise had a round-cabochon shape, while the others had a round-plate shape. All were well polished, with a smooth, clear surface, conducive to chemical and spectral investigation. Some of the samples are shown in figure 1.

## 2.2. Methods

### 2.2.1. ED-XRF

Micro-area chemical compositions were measured using an EDX-7000 energy dispersive X-ray fluorescence spectrometer, with the following test conditions: atmosphere, oxide; a voltage of 50 kV; 108 µA; 30% DT; and a 3 mm collimator. Each sample was prepared as single block, whose tested area was a circle with a diameter of 3 mm.

### 2.2.2. UV–Vis spectroscopy

Absorption spectra in the UV–Vis range were recorded using a UV-3600 UV–Vis spectrophotometer. The test conditions were described as follows: the spectral range, 300–900 nm; the scanning speed, medium; the sampling interval, 0.5 s; the scanning mode, single, reflection. Each sample was prepared as single block, with a polished, smooth surface.

### 2.2.3. Infrared spectroscopy

Infrared spectroscopy was conducted using a Tenstor 27 Fourier-transform infrared spectrometer. The measurement parameters were as follows: the wavenumber range, 2000–400 cm$^{-1}$; the voltage, 210–230 V; the frequency, 50–60 Hz; reflection method. The sample states were the same as above.

### 2.2.4. Colorimetric analysis

Turquoise colour was quantified using an X-Rite SP62 portable spectrophotometer, which collects reflective signals on the turquoise surface. The measurement conditions were as follows: the CIE standard illumination, $D_{65}$; the reflection, SCI; the observer's view, 10°; the spectral range, 400–700 nm; the measurement time, less 2.5 s; the voltage, 220 V; and the frequency, 50–60 Hz. The final colour data were averaged by three times testing. The tested area of a single sample was a circle with a diameter of 6 mm.

### 2.2.5. Uniform colour system

The colour system comprises a three-dimensional spherical colour space, with colorimetric coordinates ($a^*$ and $b^*$) in the horizontal direction and lightness ($L^*$) in the vertical direction [35]. Lightness represents a colour variation between darkest black ($L^* = 0$) and lightest white ($L^* = 100$), and a growth in lightness value means that the colour becomes brighter [38]. The colorimetric coordinate $a^*$ describes a colour variation from red ($+a^*$) to green ($-a^*$), while the colorimetric coordinate $b^*$ describes a colour variation from yellow ($+b^*$) to blue ($-b^*$) [36]. Chroma ($C^*$) represents a saturation variation of single colour (e.g. blue), between lightest blue ($C^* = 0$) and deepest blue ($C^* = 100$). A colour with high chroma is more brilliant and intensive [38]. Hue angle ($h^o$) varies from 0 to $2\pi$, representing a series of continuous colour variation from red, yellow, green, blue to purple [36]. All the colour parameters are psycho-physical parameters without unit. $C^*$ and $h^o$ can be calculated using $a^*$ and $b^*$ as follows:

$$C^* = \sqrt{a^{*2} + b^{*2}} \tag{2.1}$$

and

$$h^o = \arctan\frac{b^*}{a^*}. \tag{2.2}$$

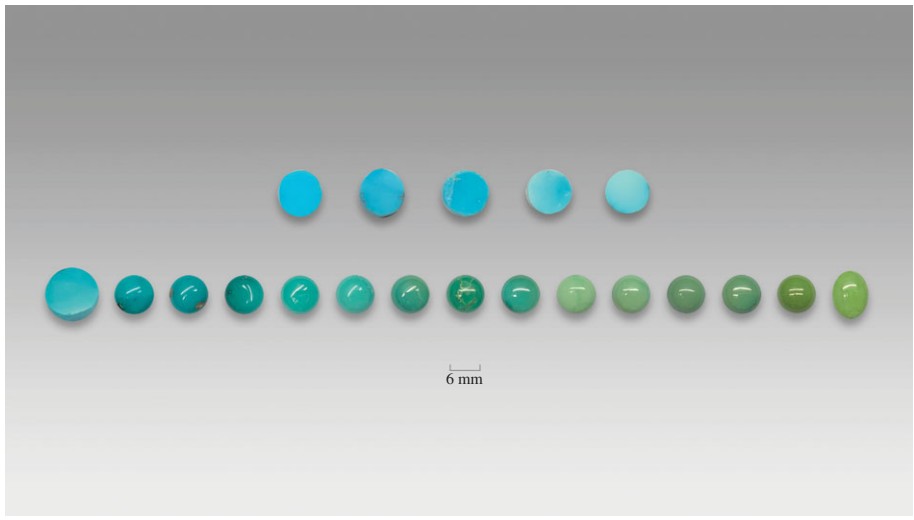

**Figure 1.** The top turquoise samples possess pure blue with slight colour variation. The bottom 15 samples display a continuous colour transition from blue to yellow-green.

To calculate the colour difference in turquoise caused by increasing absorbed-water content, we use the CIE DE2000 ($\Delta E_{00}$) formula; this improves visual uniformity to compare with CIE $L^*a^*b^*$ [50]. $\Delta E_{00}$ calculates colour differences more precisely in the green-blue area and the formula is as follows:

$$\Delta E_{00} = \sqrt{\left(\frac{\Delta L^*}{K_L S_L}\right)^2 + \left(\frac{\Delta C^*}{K_C S_C}\right)^2 + \left(\frac{\Delta H^*}{K_H S_H}\right)^2 + R_T \left(\frac{\Delta C^*}{K_C S_C}\right)^2 \left(\frac{\Delta H^*}{K_H S_H}\right)}, \tag{2.3}$$

where $\Delta L^*$, $\Delta C^*$ and $\Delta H^*$ are the differences in lightness, chroma and hue angle. $R_T$ is a conversion function that reduces the interaction between chroma and hue angle in the blue area. $S_L$, $S_C$ and $S_H$ are functions that calibrate for the absence of visual uniformity in the CIE $L^*a^*b^*$ formula. $K_L$, $K_C$ and $K_H$ are correction parameters for the experimental environment ($K_L = 1$, $K_C = 1$, $K_H = 1$). The CIE DE2000 (1 : 1 : 1) was chosen, as it calculates more accurately colour difference.

# 3. Results and discussion

## 3.1. Chemical composition analysis

ED-XRF can be used to quickly detect the content of most elements in turquoise. A quantitative analysis of the chemical composition of 35 turquoise samples based on FP standard curve shows turquoise oxide content to be the following (table 1): $w(P_2O_5) \in (32.827, 45.547)$, $w(Al_2O_3) \in (22.100, 32.437)$, $w(CuO) \in (1.403, 22.491)$, $w(Fe_2O_3) \in (0.621, 15.192)$, $w(ZnO) \in (0.089, 8.296)$, where $w(Fe_2O_3)$ represents the total iron content detected in turquoise. The blue turquoise is rich in copper, whose $w(CuO)$ ranges from 18.530% to 22.491% with an average of 20.008%. Fe-rich and Zn-rich samples are identified as 'chalcosiderite' and 'faustite', respectively.

Turquoise is characterized by a triclinic structure (figure 2). Various site nomenclatures have been used to investigate the crystal structure and spectra of the turquoise group, dependent on chemical compositions. Here, we use a more general site nomenclature to explain the position of different metal cations, avoiding phrases such as 'Fe$^{3+}$ occupying an Al site' [8]. The turquoise group structurally consists of distorted XO6 octahedra, MO6 octahedra and PO4 tetrahedra, where the X site is occupied by medium-sized divalent cations (e.g. $Cu^{2+}$ and $Zn^{2+}$) and the P site is occupied by phosphorus. The MO6 octahedra have three sites (M1–M3) occupied by small trivalent cations (primarily Al$^{3+}$, Fe$^{3+}$ and V$^{3+}$). Therefore, the turquoise group is also expressed as $X(M_1M_2M_3)_6(PO_4)_4(OH)_8 \cdot 4H_2O$ (X = A, M = B). Pairs of edge-sharing M1 and M2 octahedra are linked through sharing $O^{2-}$ corners with pairs of PO4 tetrahedra to form chains extending along the $b$-direction. These chains are linked in the $a$- and $c$-directions through sharing $O^{2-}$ corners with M3 octahedra to further combine with X octahedra sharing edges with the M1 and M2 octahedra. Non-edge sharing with M1 and M2

**Table 1.** The ED-XRF data of turquoise samples.

| sample | B11 | B16 | B18 | G01 | G04 | G06 | G07 | G12 | G09 | G14 |
|---|---|---|---|---|---|---|---|---|---|---|
| $h°$ | 223.60 | 220.18 | 215.99 | 191.42 | 180.98 | 167.17 | 158.64 | 140.79 | 137.54 | 121.32 |
| wt(%)/oxides | | | | | | | | | | |
| $P_2O_5$ | 44.119 | 44.263 | 45.121 | 37.827 | 42.210 | 42.458 | 40.799 | 40.451 | 42.982 | 43.586 |
| $Al_2O_3$ | 32.092 | 31.558 | 31.322 | 22.100 | 41.765 | 40.657 | 35.239 | 35.465 | 42.874 | 42.635 |
| $CuO$ | 20.313 | 19.638 | 19.599 | 21.026 | 10.620 | 9.608 | 9.009 | 7.850 | 4.062 | 1.403 |
| $Fe_2O_3$ | 0.695 | 1.276 | 1.589 | 15.192 | 3.660 | 4.936 | 9.010 | 11.874 | 0.621 | 0.780 |
| $ZnO$ | 0.361 | 0.588 | 0.529 | 0.329 | 0.187 | 0.127 | 1.544 | 0.216 | 6.219 | 8.296 |
| $V_2O_5$ | 0.035 | — | — | 0.264 | 0.183 | 0.223 | 0.501 | 0.396 | 0.448 | 1.649 |
| $Cr_2O_3$ | 0.030 | 0.052 | 0.049 | 0.033 | 0.359 | 0.351 | 0.792 | 0.629 | 0.526 | 0.419 |
| others | 2.355 | 2.625 | 1.791 | 3.658 | 1.016 | 1.570 | 3.106 | 3.119 | 2.268 | 1.239 |
| total | 100.00 | 100.00 | 100.00 | 100.00 | 100.00 | 100.00 | 100.00 | 100.00 | 100.00 | 100.00 |

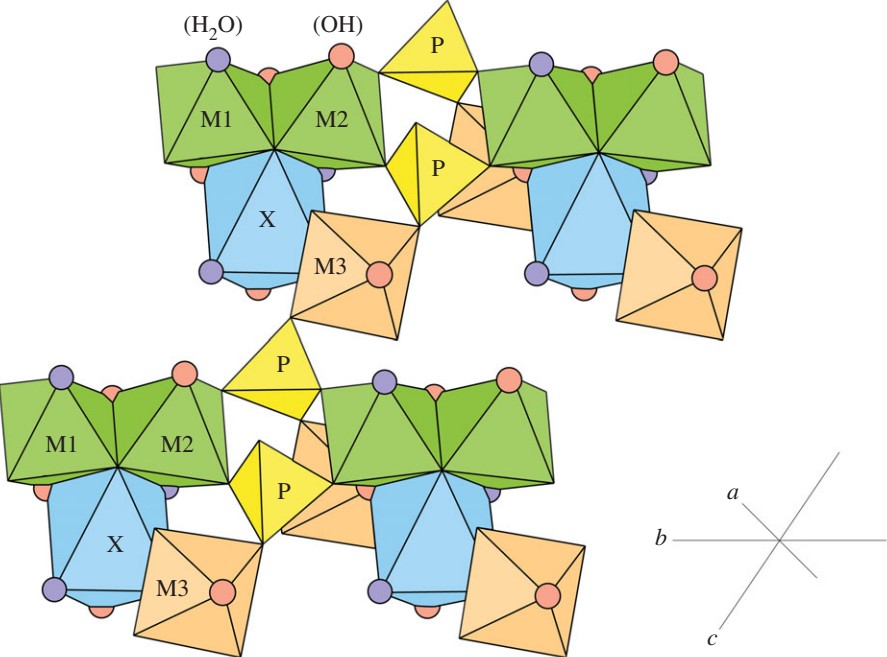

**Figure 2.** The crystal structure of turquoise; X octahedra are shown in blue, P tetrahedra are shown in yellow, M1 and M2 octahedra are shown in green and M3 octahedra are shown in orange; hydrogen bonds are omitted for clarity. OH and $H_2O$ groups are shown as red and violet circles, respectively (referring from Abdu *et al.* [8]).

octahedra results in difference in the local stereochemistry of the M3 octahedra. The heteropolyhedral framework is further strengthened by hydrogen bonds networks in three dimensions [7].

To analyse the chemical composition of turquoise with various colours, 35 turquoise samples were divided into six groups based on five centres of $h°$ (105°/135°/165°/195°/225°) [52], including blue $h° \in$ (215.99, 223.70), greenish blue $h° \in$ (187.99, 192.59), bluish green $h° \in$ (177.45, 180.98), green $h° \in$ (158.64,169.39), yellowish green $h° \in$ (137.54, 140.79) and greenish yellow $h° \in$ (116.47, 130.45). Based on colour nomenclature, we know that blue-green includes greenish blue and bluish green, and yellow-green includes yellowish green and greenish yellow. Cu content, Fe content and Zn content are analysed by box chart (figure 3a), where the blue group is high in Cu and extremely low in Fe and Zn. The w(CuO) in green turquoise is significantly lower than that in blue turquoise, which varies slightly, reaching its lowest value in greenish yellow faustite. There exists a high Fe content in green and yellowish green turquoise, which gradually decreases with increasing blue and yellow hue. Because Zn is a rare element, only faustite is rich in Zn, with w(ZnO) > w(CuO), negatively correlated with the hue angle of turquoise. The results indicate that $w(Fe_2O_3)$ is poor correlated with w(CuO) in all samples, indicating no obvious substitution of Cu by $Fe^{2+}$ in the crystal structure of turquoise.

Fe-rich turquoise has a green hue. The empirical expression between Fe content and Al content is as follows (figure 3b):

$$w(Fe_2O_3) = 52.647 − 1.175w(Al_2O_3) \quad (r = −0.990 \quad R^2 = 0.980), \tag{3.1}$$

where $r$ represents Pearson's correlation coefficient, describing the correlation between two variables. $R^2$ represents the determination coefficient, reflecting the variability percentage of the dependent variable. The results indicate that $w(Fe_2O_3)$ is significantly negatively correlated with $w(Al_2O_3)$, which means that substitution of Al by $Fe^{3+}$ results in the solid-solution series turquoise-chalcosiderite: Cu(Al, $Fe^{3+})_6(PO_4)_4(OH)_8 \cdot 4H_2O$. However, $w(Cr_2O_3)$ and $w(V_2O_5)$ are detected in green-hue turquoise, which may also result in a decreasing Al content.

Partial correlation analysis is effective in analysing the correlations among different variables when taking control of a relative variable. $Fe^{3+}$, $V^{3+}$ and $Cr^{3+}$ may replace with $Al^{3+}$ in the crystal structure of turquoise. Since $w(V_2O_5)$ and $w(Cr_2O_3)$ are extraordinarily lower than $w(Fe_2O_3)$, we choose $w(Fe_2O_3)$ as a control variable to examine the correlation between Cr–V content and Al content. A partial correlation analysis of Al–V–Cr content shows that both $w(V_2O_5)$ and $w(Cr_2O_3)$ have a good partial correlation with $w(Al_2O_3)$, which explains the discrete grey points in figure 3b. The substitution

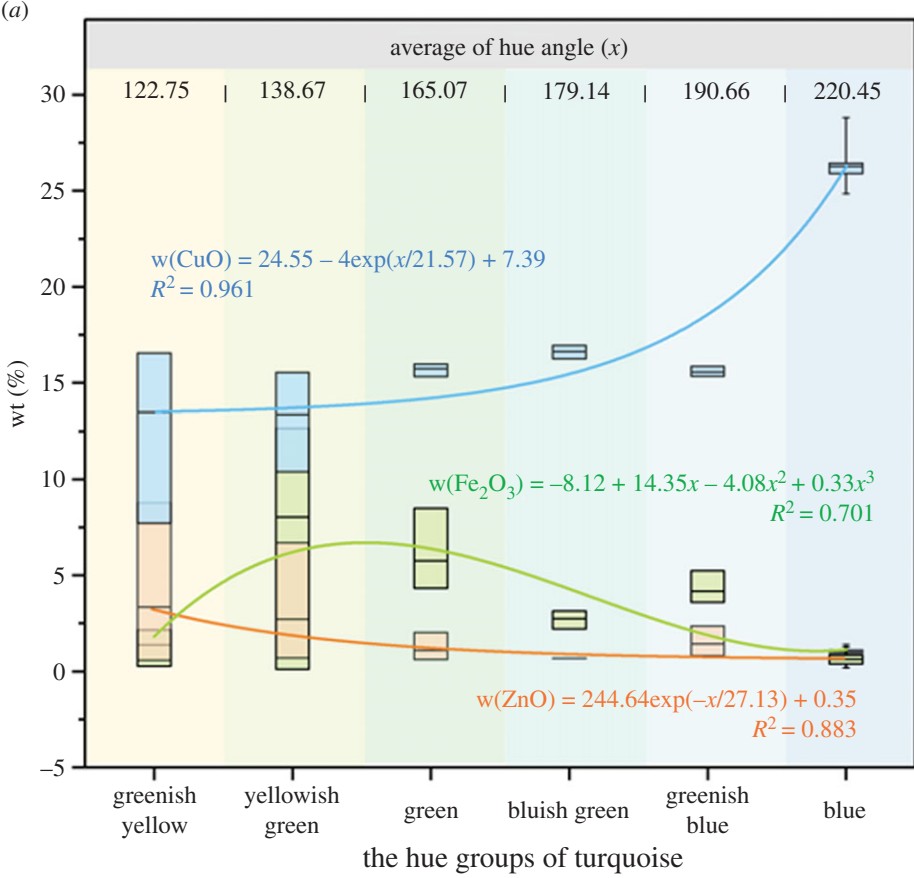

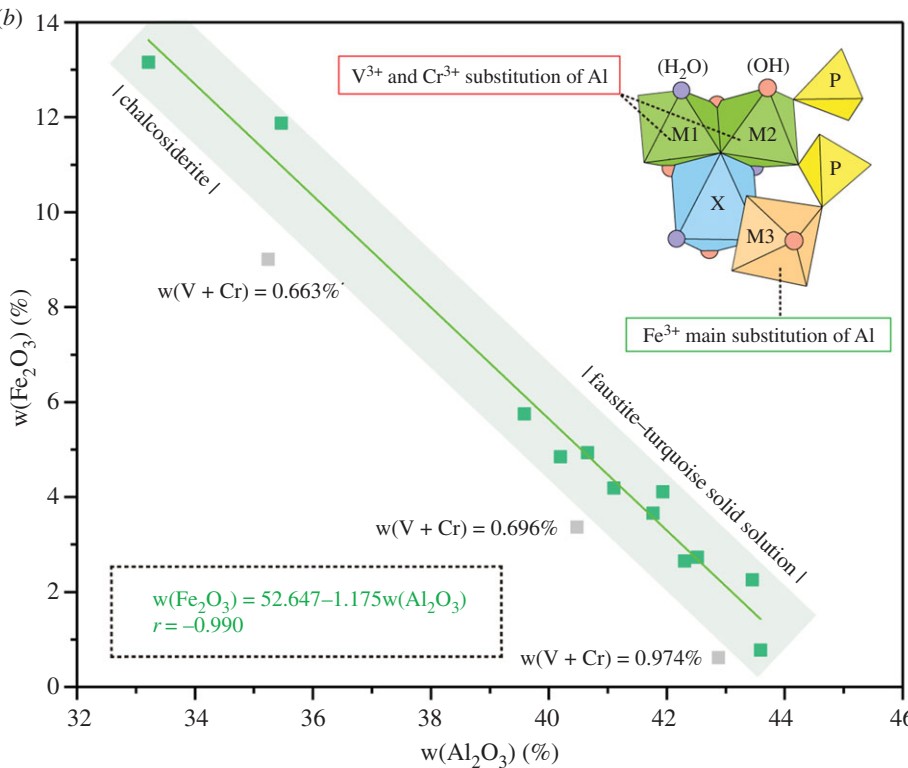

**Figure 3.** (a) Box chart of Cu content, Fe content and Zn content in turquoise (classified in six hue groups). w(CuO) is shown in blue, w($Fe_2O_3$) is shown in green and w(ZnO) is shown in orange. (b) w($Al_2O_3$) versus w($Fe_2O_3$) of the ED-XRF data in the green-hue turquoise, where the discrete grey points are due to the substitution of Al by $V^{3+}$ and $Cr^{3+}$.

of Al by $V^{3+}$ and $Cr^{3+}$ results in lower value of $w(Fe_2O_3)/w(Al_2O_3)$. The trivalent metal cations, including $Fe^{3+}$, $V^{3+}$ and $Cr^{3+}$, show representative absorption in the UV–Vis spectra, which contributes to the green hue in turquoise.

## 3.2. UV–Vis spectra

UV–Vis reflective spectra can accurately illustrate absorption characteristics of transition metal cations in turquoise. For Cu-rich turquoise (figure 4a), the absorption band in the orange-red region occurs around 673 nm due to the $d$–$d$ electron transition of $Cu^{2+}$ [6], and it can extend to 800 nm, as it combines with the weak absorption band caused by $Fe^{2+}$. In addition, the double absorption peaks at 422 and 428 nm in the violet-blue region are caused by the electron transition of $Fe^{3+}$ ($^6A_1 \rightarrow {}^4E$ and $^4A_1(^4G)$), while the weak absorption band in the ultraviolet region of 370 nm is caused by the $Fe^{3+}$ electron transition ($^6A_1 \rightarrow {}^4E(^4D)$) and charge transferring from $O^{2-}$ to $Fe^{3+}$ [17]. Increasing Fe content leads to an enlargement of the peak area at 428 nm and an enhancement in the resolution of the double peaks.

For turquoise and chalcosiderite, the double peaks at 422 and 428 nm finally merge into a strong narrow band at 426 nm, reflecting a bathochromic shift from 426 to 428 nm before decreasing in absorption strength with increasing V content (figure 4b). The samples undergo a colour transition from blue to green in the process. The absorption band near 370 nm broadens, reflecting a bathochromic shift with increasing Fe content. The weak band in the blue region at 470–480 nm is due to the special $Fe^{3+}$ lattice position [6], contributing to the yellow hue of turquoise. The results indicate that Fe and V trivalent cations can work together to enhance absorption in the violet and blue-violet regions, and are responsible for a blue-to-green transition.

Overall, the absorption reflectivity at 428 nm is negatively correlated with the lightness of turquoise and has no obvious correlation with Fe content, indicating that turquoise with high lightness has high reflectivity at 428 nm peak, which can be expressed as

$$R_{428nm} = 2.171L^* - 120.031 \quad (r = 0.937). \tag{3.2}$$

For faustite, there is no obvious absorption arising from Zn in the UV–Vis reflective spectra. The substitution of Cu by $Zn^{2+}$ results in a decrease in Cu content and reduces absorption in the orange-red region to produce a hypochromic effect on blue. The electron transition of $V^{3+}$ produces two absorption bands in the orange (620 nm) and violet-blue regions (420–460 nm) [13]. The band in the violet-blue region completely covers the absorption peaks at 422 and 428 nm derived from the Fe-electron transition, and broadens with increasing V content. Small levels of Cr in faustite can generate a narrow absorption band in the green region of 568 nm, and work with Cu to produce a broad absorption band in the red region of 683 nm [14]. The results indicate that $Zn^{2+}$ suppresses the colour blue in turquoise, $V^{3+}$ enhances absorption in the violet, blue and orange regions, and $Cr^{3+}$ enhances absorption in the red and green regions, all of which works together to form the vivid greenish yellow in faustite (figure 5).

A stepwise regression method is often used to screen out variables with significant correlation and to judge the contributions of independent variables to dependent variables [53]. Therefore, we used this method to study the contributions of colour-causing metal elements (Cu, Fe, Zn, V and Cr) to $b^*$ (the colorimetric coordinate) in turquoise and to explain which element more sensitively influences the hue variation, from blue to yellow in turquoise. The results show that the determination coefficient ($R^2$) in the $b^*$–Cu–Fe model is 0.886 with $w(ZnO)$, $w(V_2O_5)$ and $w(Cr_2O_3)$ as excluded variables, indicating that copper and iron contents have good correlation with $b^*$ (table 2). The significance coefficient is lower than 0.001, indicating a good linear regression. Thus, Cu and Fe content mainly contributes to the hue variation from blue to yellow in turquoise.

## 3.3. FTIR

The infrared spectra in 2000–400 $cm^{-1}$ range possess prominent absorption bands arising from the vibrations of the OH, $H_2O$ and $(PO_4)^{3-}$ units (table 3). The P–O stretching vibrations of the phosphate units $(PO_4)^{3-}$ are located at 1200–900 $cm^{-1}$, while the coupled motions of the tetrahedral and octahedral frameworks are situated below 700 $cm^{-1}$ [8]. The O–H bending vibrations of the OH units are located at 833 and 781 $cm^{-1}$, while that of the $H_2O$ molecules are located near 1637 $cm^{-1}$ [30,31]. Since the M–$H_2O$ has weak hydrogen bonds, the infrared spectra of turquoise shows weak absorption when the wavenumber is over 1200 $cm^{-1}$ [32]. The C–O stretching vibrations of the carbonate group $(CO_3)^{2-}$ are located near 1450 $cm^{-1}$, with weak absorption, which can possibly be attributed to the

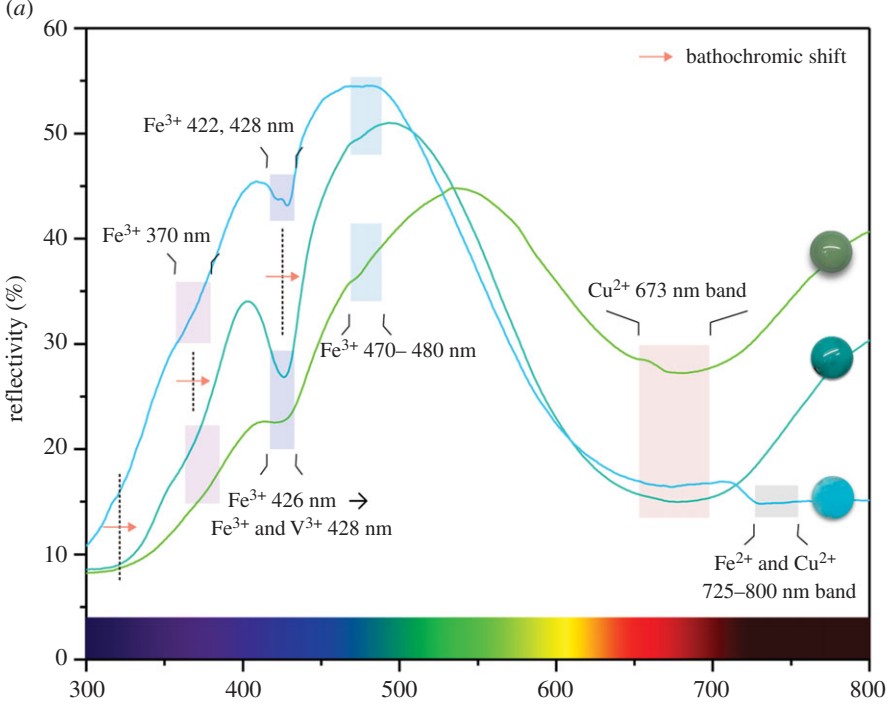

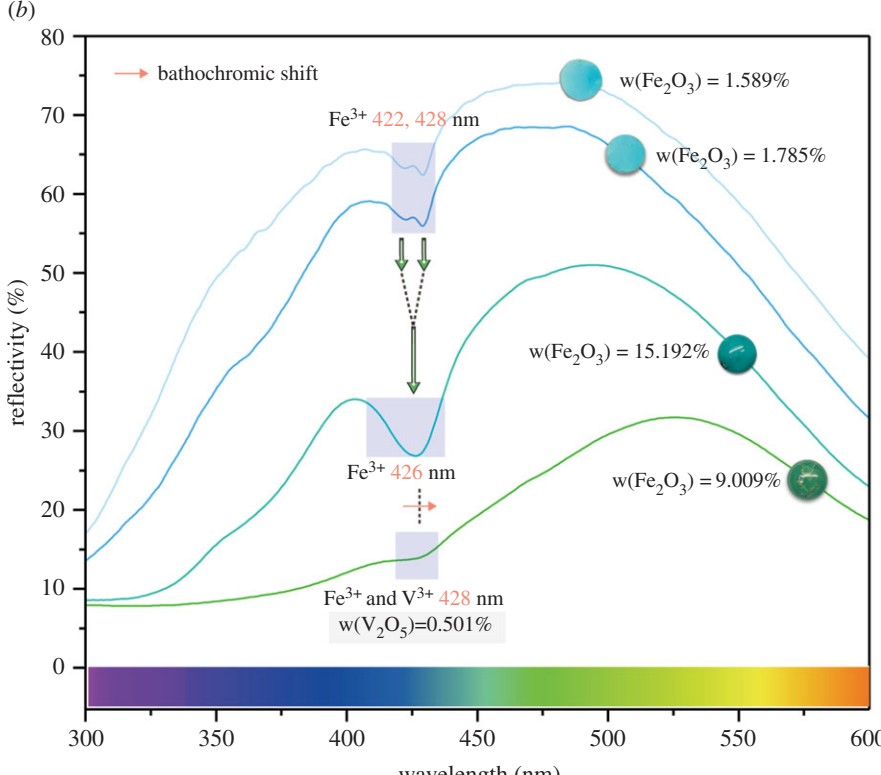

**Figure 4.** (*a*) UV–Vis spectra of the solid-solution series turquoise-chalcosiderite, where the red arrow means that the absorption band has a bathochromatic shift. (*b*) UV–Vis spectra of turquoise in the 300–600 nm region, reflecting that the double absorption peaks merge into single peak at 426 nm, which shifts to 428 nm, with increasing V content.

presence of the admixed carbonate on the chalcosiderite-faustite surface [54]. Different metal cations can cause changes in frequency and absorption strength.

Cu-rich turquoise clearly has a lower reflectivity than Fe-rich chalcosiderite and Zn-rich faustite, especially in the spectral region caused by the phosphate group $(PO_4)^{3-}$ (figure 6*a*). $PO_4$ tetrahedra in

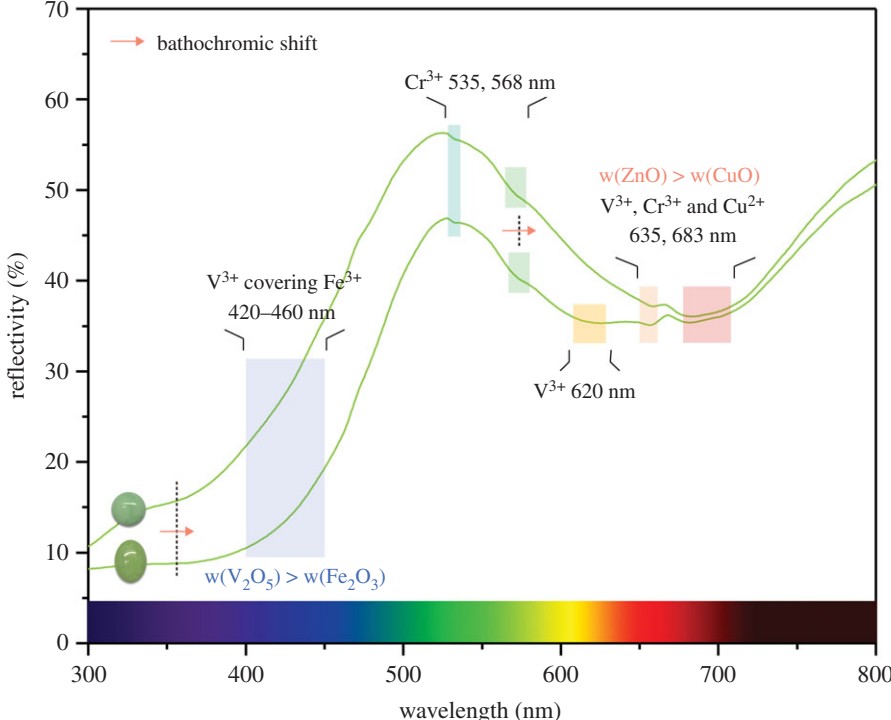

**Figure 5.** UV–Vis spectra of faustite, where the red arrow means that the absorption band has a bathochromatic shift.

**Table 2.** Model of $b^*$–Zn–Fe stepwise regression.

| model | factors | unstandardized | standardized | $T$ | sig | $R^2$ | adjusted $R^2$ |
|---|---|---|---|---|---|---|---|
| 1 | constant | 29.094 | — | 9.999 | 0.000 | 0.857 | 0.853 |
| | CuO (%) | −2.477 | −0.926 | −14.064 | 0.000 | | |
| 2 | constant | 24.126 | — | 7.615 | 0.000 | 0.886 | 0.879 |
| | CuO (%) | −2.310 | −0.863 | −13.565 | 0.000 | | |
| | Fe$_2$O$_3$ (%) | 0.777 | 0.181 | 2.839 | 0.008 | | |

the crystal structure of turquoise shares $O^{2-}$ corners with MO6 octahedra, and small-sized trivalent cations ($Al^{3+}$) form strong bonds with $O^{2-}$ ions, resulting in a weak polarity in the phosphate groups $(PO_4)^{3-}$ in blue turquoise. Substitutions of Al by medium-sized trivalent cations ($Fe^{3+}$, $V^{3+}$ and $Cr^{3+}$) change the lattice environment of chalcosiderite and faustite, where the bond energy of Fe–O is lower than that of Al–O, enhancing the polarity of the phosphate group $(PO_4)^{3-}$ and leading to high reflectivity in the infrared spectra. Increasing Fe content reduces the peak resolution of P–O stretching vibration below 700 cm$^{-1}$, and chalcosiderite loses absorption peaks at 609 and 648 cm$^{-1}$, distinguishing from turquoise. The absorption peak at 1057 cm$^{-1}$ in turquoise is separated into two peaks at 1061 and 1047 cm$^{-1}$ in faustite.

Previous studies have suggested that the OH bending variation of the OH units can be used to determine the origin of turquoise [30–32], and its absorption peaks have been used to identify differences in the chemical composition of turquoise. Therefore, we focused on OH bending variation and carried out K-means clustering analysis and Fisher discriminant analysis to divide all turquoise samples into different groups based on their chemical composition.

K-means clustering analysis allows the quick division of research objects into relatively homogeneous groups [55–57]. Fisher discriminant analysis, one of the most important multivariate statistical analysis methods, can summarize the common feature from various groups to obtain discriminant formulae, identifying the accuracy of clustering analysis [58]. The clustering significance (sig) is lower than 0.001 with a cluster number of 3, as w(CuO), w(Fe$_2$O$_3$) and w(ZnO) are independent variables.

**Table 3.** Infrared peaks of turquoise samples.

| | turquoise | | chalcosiderite | | faustite | |
|---|---|---|---|---|---|---|
| | B11 | B16 | G01 | G12 | G09 | G14 |
| H–O–H bending vibration | 1637 | 1637 | — | — | — | — |
| C–O stretching vibration | — | — | 1448 | 1448 | 1450 | 1450 |
| P–O stretching vibration | 1199 | 1196 | — | — | — | — |
| | 1121 | 1123 | 1118 | 1117 | 1119 | 1121 |
| | 1059 | 1059 | 1051 | 1057 | 1061/1043 | 1061/1043 |
| | 1007 | 1007 | 1009 | 1009 | 1005 | 1003 |
| O–H bending vibration | 833 | 833 | 833 | 833 | 831 | 798 |
| | 781 | 781 | 781 | 781 | 796 | |
| P–O stretching vibration | 650 | 650 | 646 | — | 648 | 646 |
| | 608 | 608 | — | — | 623 | 623 |
| | 571 | 569 | 565 | 567 | 588 | 590 |
| | 536 | 534 | — | 536 | 536 | 553 |
| | 484 | 480 | 478 | 480 | 480 | 480 |
| | 449 | 451 | 449 | 451 | 453 | 453 |

All sigs in the Fisher discriminant analysis are lower than 0.001, indicating that discriminant variables can work well in the classification. The discriminant formulae are shown in table 4 with an accuracy as high as 100.00%, suggesting that the model is reliable for the classification results of the chemical composition in turquoise. Thus, all samples are classified into three categories, matching three end-members in the turquoise group, including turquoise (21 pieces), chalcosiderite (12 pieces) and faustite (two pieces).

The $R_{OH}$ representing the reflectivity ratio of the double peaks at 781 and 833 cm$^{-1}$ is negatively correlated with Cu content and is used to identify the three turquoise categories (figure 6b). Chalcosiderite has a higher resolution and absorption reflectivity than turquoise and faustite in the 700–900 cm$^{-1}$ range, and its $R_{OH}$ ranges from 0.765 to 0.885, with an average value of 0.802. The $R_{OH}$ of faustite is over 1.000, which allows faustite to be distinguished from other subspecies in the turquoise group. However, greenish yellow faustite is high in V and shows a single peak at 798 cm$^{-1}$, with w(V$_2$O$_5$) > 1.00%, lacking the double peaks from OH bending vibration.

## 3.4. Water immersion experiment

Adsorbed water content is one of the most important factors that influence turquoise colour, and one which influences turquoise texture and crystallinity. A water immersion experiment was designed in this paper. Thirty-five samples of turquoise were measured using the portable spectrophotometer to obtain control colour data before the experiment. They were then immersed in clean water under atmospheric temperature and pressure conditions for 24 h. They were removed from the water, wiped away free water on the surface and were quickly measured again to obtain final colour data. Specific gravity (SG) testing and scanning electron microscopy (SEM) experiment were carried out on the samples before water immersion experiment to better elucidate structural and textural characteristics of turquoise with different SGs.

### 3.4.1. Micro-observation analysis

The SG of turquoise was measured using a hydrostatic weighing scale with an accuracy of 0.001 g, and samples were put in water for less than 2 s. The SG of turquoise ranges from 2.59 to 2.87 based on the

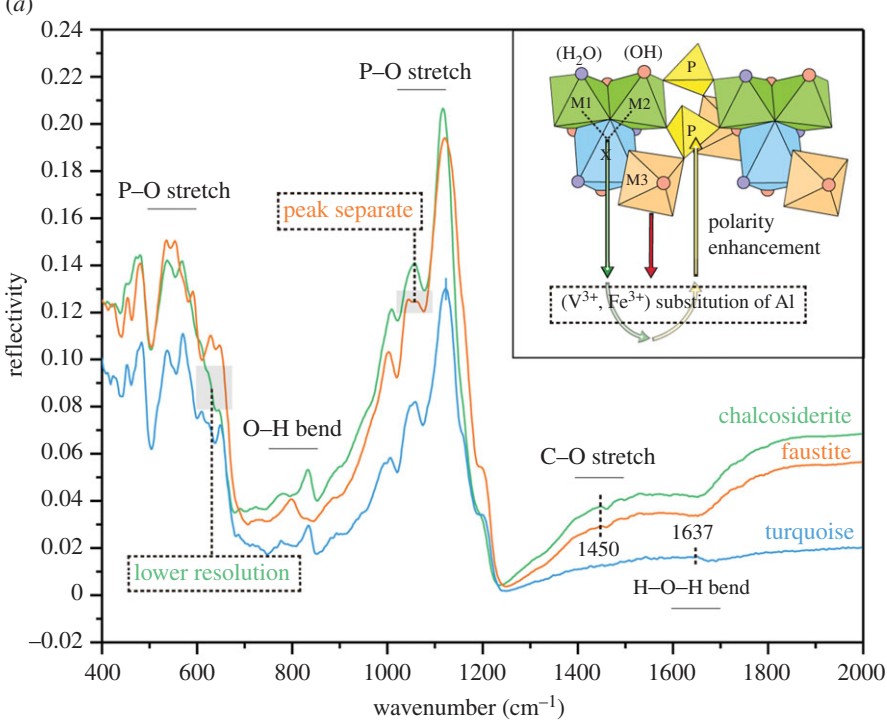

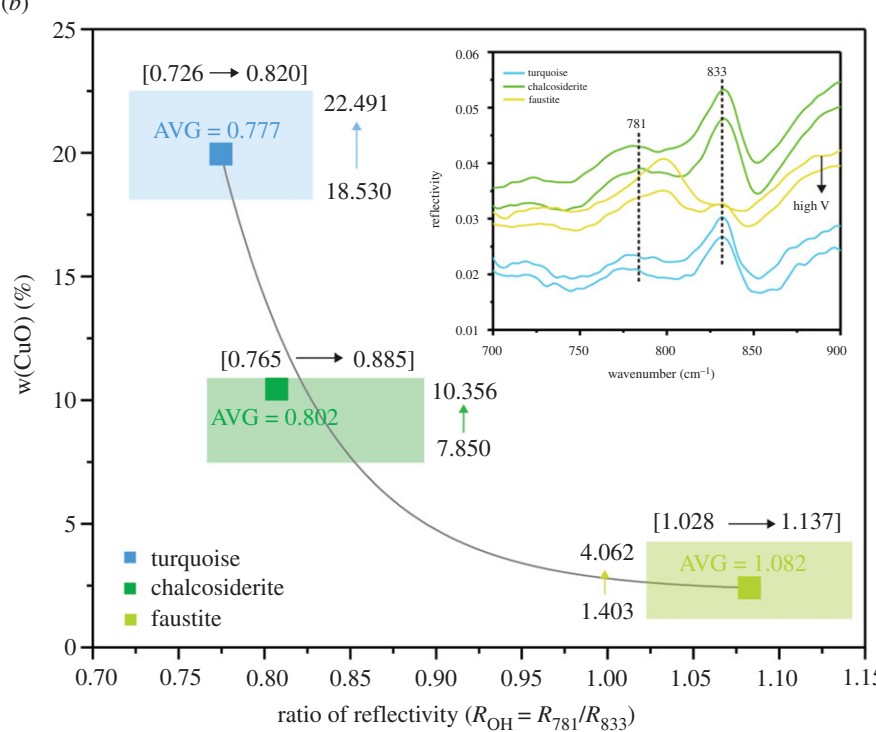

**Figure 6.** (a) FTIR spectra of turquoise, chalcosiderite and faustite. (b) Correlation between $R_{OH}$ and Cu content. The $R_{OH}$ data presents in three-group distributions on the diagram, based on K-means clustering analysis and Fisher discriminant analysis; turquoise is shown in blue, chalcosiderite is shown in green and faustite is shown in green-yellow. FTIR spectra of them in the 700–900 cm$^{-1}$ region reflect changes in frequency and absorption strength.

Archimedes principle. Two samples (B05 and B06) with different SGs (2.72 and 2.66) were selected for SEM analysis. Scanning electron microscopy was used to examine the micro-structure and morphology of turquoise surface, especially with respect to its structural compactness and porosity [8,59,60].

Detailed petrography of the SEM images shows distinct differences between loose and dense turquoise. The turquoise (B05) with a high SG has a relatively homogeneous texture and blocky

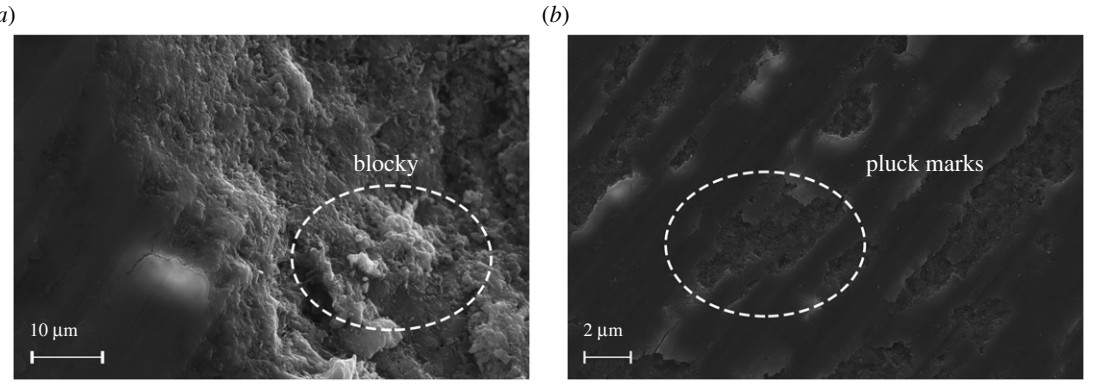

**Figure 7.** SEM images of moderate-dense turquoise (sample B05, SG = 2.72) (*a*) with a crystalline blocky morphology and (*b*) a few pluck marks caused by polishing.

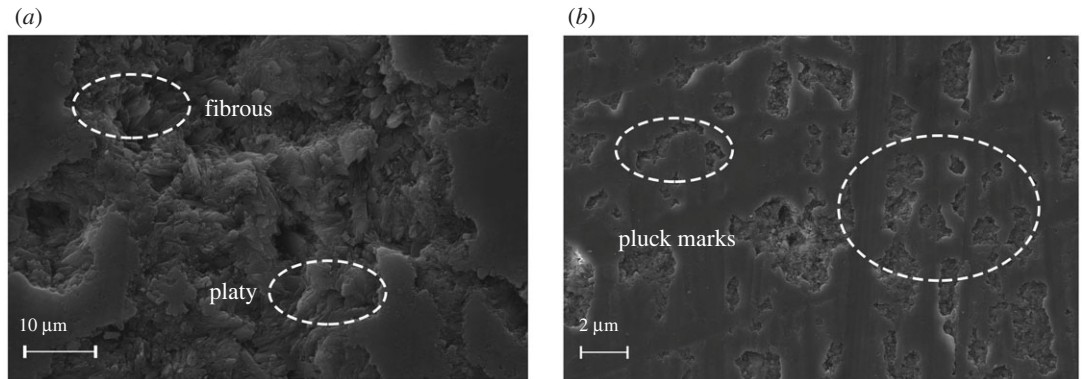

**Figure 8.** SEM images of low-dense turquoise (sample B06, SG = 2.66) (*a*) with a fibrous and platy morphology and (*b*) more pluck marks than the sample B05.

**Table 4.** Fisher discriminant accuracy.

| (%) | 1 | 2 | 3 | numbers | the discriminant formulae |
|---|---|---|---|---|---|
| 1 | 100.0 | 0.0 | 0.0 | 21 | $F_1 = 5.645w(CuO) - 1.788w(Fe_2O_3) + 9.426w(ZnO) - 59.045$ |
| 2 | 0.0 | 100.0 | 0.0 | 2 | $F_2 = 3.130w(CuO) - 1.576w(Fe_2O_3) + 39.237w(ZnO) - 147.205$ |
| 3 | 0.0 | 0.0 | 100.0 | 12 | $F_3 = 2.470w(CuO) + 0.023w(Fe_2O_3) + 4.666w(ZnO) - 15.169$ |

morphology, with a few pluck marks as voids due to polishing (figure 7). However, the turquoise (B06) with a low SG shows a porous texture, consisting of fibrous and platy crystallites, and it has more pluck marks than the sample B06 (figure 8).

### 3.4.2. Colour variation analysis

Previous research using differential thermal thermogravimetric analysis confirms that the absorbed water content, easily escaping from turquoise pores or textural fractures, is less than 2% and most of the turquoise cannot be water-saturated without human intervention. According to naked-eye observations during water immersion experiment, all samples displayed significant colour variation in the first 1 min, which then became steady with no further naked-eye colour change after 5 min; turquoise colour changed intensive after the experiment, which was more obvious in blue turquoise compared with samples of different colours. A weaker colour difference was found in denser turquoise samples. The intense colour change occurred in loose turquoise samples that had air bubbles on their surface.

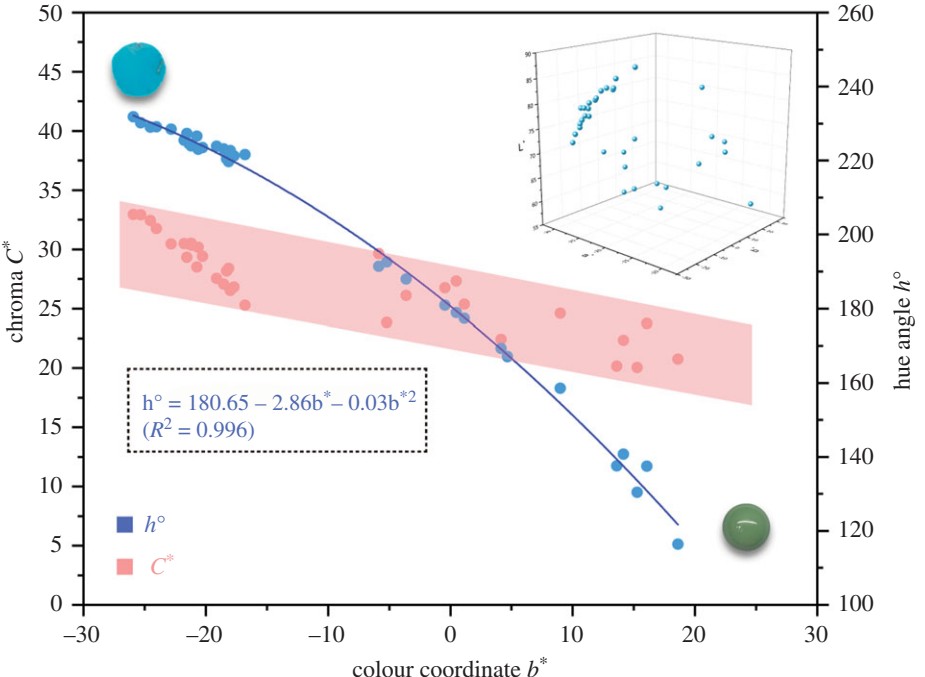

**Figure 9.** Relationship between colorimetric coordinate $b^*$ and hue angle $h^\circ$ or chroma $C^*$ in turquoise's colour. The colour plots of turquoise distribute in the uniform colour space CIE 1976 $L^*a^*b^*$ on the top right. Note: $a^* \in (-29.06, -9.26)$, $b^* \in (-25.27, 26.80)$ and $L^* \in (55.86, 88.31)$.

Based on the CIE 1976 $L^*a^*b^*$ colour system, the control colour data of all turquoise samples before water immersion experiment is noted: lightness $L^* \in (55.86, 85.26)$, chroma $C^* \in (20.07, 32.95)$ and hue angle $h^\circ \in (116.47, 231.86)$. An empirical expression shows that hue angle of turquoise has extremely negative correlation with colorimetric coordinate $b^*$ $(x)$ (figure 9)

$$h^o = 180.65 - 2.68x - 0.03x^2 \quad (R^2 = 0.996). \tag{3.3}$$

The $R^2$ value is 0.996, indicating that $b^*$ can explain the variation in hue angle for most turquoise.

The chromatic diagram (a two-dimensional plane composed of coordinates $a^*$ and $b^*$) shows that the $a^*$–$b^*$ range of blue turquoise becomes narrower after water immersion experiment and shifts to the negative direction of coordinate axis $b^*$, indicating that increasing absorbed-water content enhances blue hue and makes colour more brilliant. However, the $a^*$–$b^*$ range of blue-green turquoise slightly shifts to the positive direction of the coordinate axis $a^*$, indicating that increasing absorbed-water content enhances green hue. For yellow-green turquoise, the $a^*$–$b^*$ range becomes narrower in the direction of coordinate axis $a^*$, indicating that the yellow-green turquoises show low colour resolution between individuals after water immersion experiment (figure 10a).

Based on the six hue groups mentioned above, further quantitative analysis was carried out to investigate the impact increased absorbed-water content had on the lightness difference ($\Delta L^*$), chroma difference ($\Delta C^*$) and hue angle difference ($\Delta h^\circ$) (figure 10b). The results show that an increase in absorbed-water content results in a decrease in lightness of colour for all turquoise samples ($\Delta L^* < 0$); this indicates that absorbed water has a negative effect on the turquoise lightness. The negative effect of adsorbed water on greenish yellow turquoise is the most significant with $|\Delta L^*|$ maximum as high as 14.39. The blue turquoise displays intense colour and enhances chroma with increasing absorbed-water content; this indicates that absorbed water has a positive impact on the chroma ($\Delta C^* > 0$, $|\Delta C^*|$ max = 9.52), while yellow-green turquoise appears deeper in colour with decreasing in chroma. However, blue-green turquoise shows extremely low chroma differences, which is difficult to attribute to changes in absorbed-water content or instrument error. The hue angle of blue and yellow-green turquoises is positively affected by absorbed water ($\Delta h^\circ > 0$) while that of blue-green turquoise is negatively affected.

(*a*)

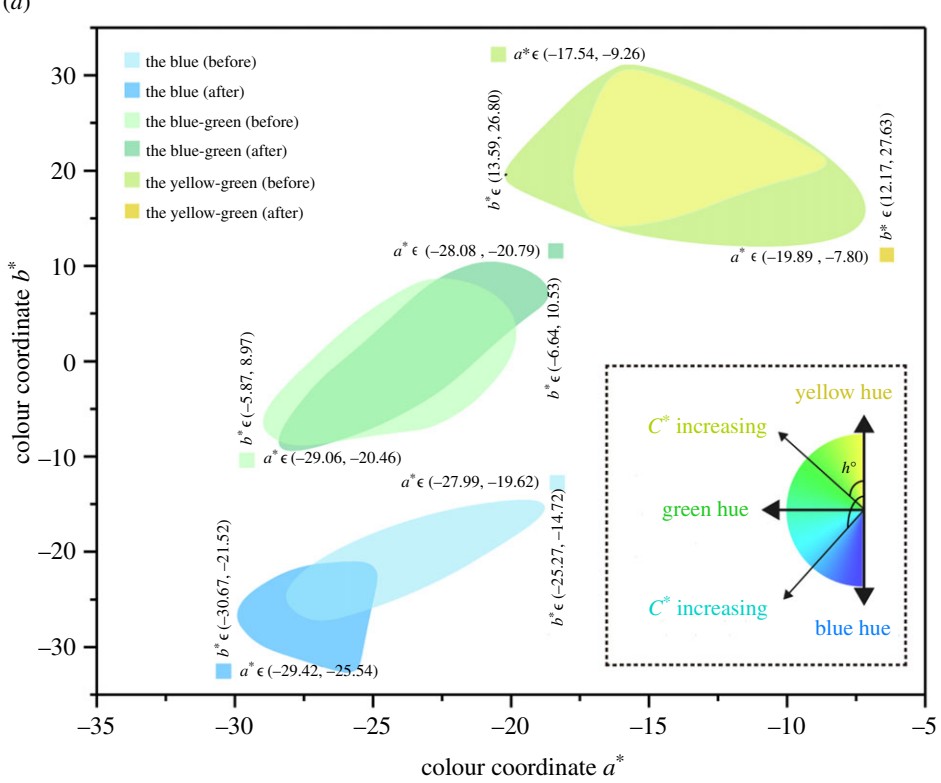

(*b*)

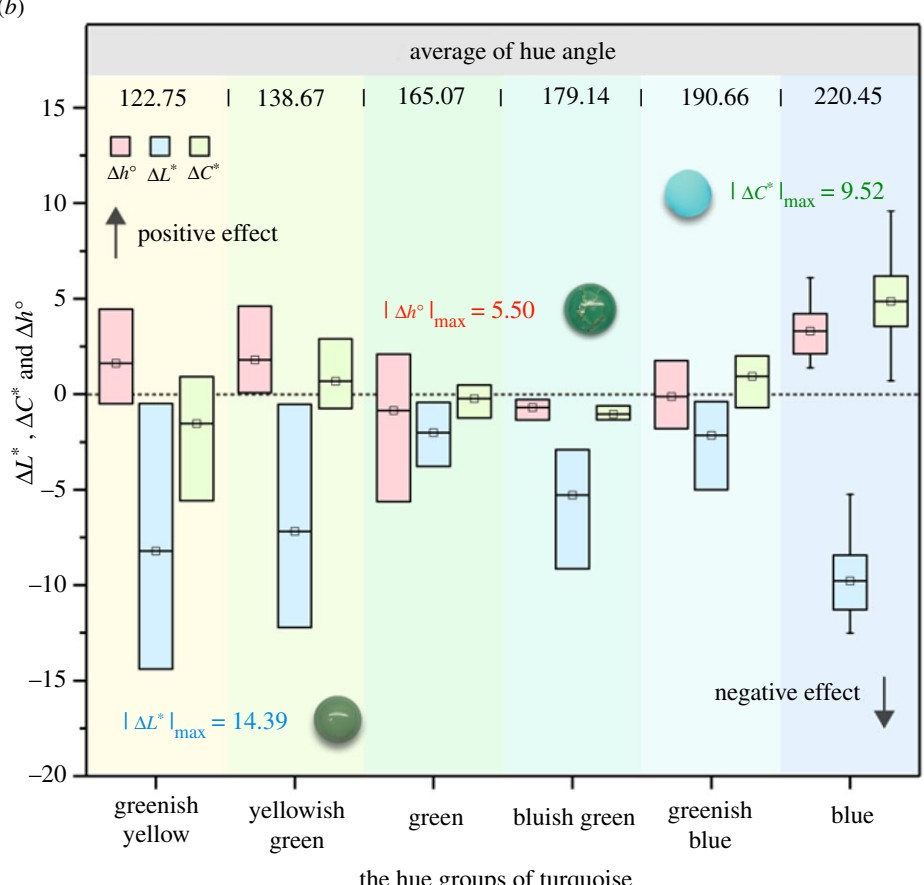

**Figure 10.** (*a*) CIE 1976 $L^*a^*b^*$ chromatic diagram of turquoise colours before and after water immersion experiment. (*b*) Box chart of lightness difference ($\Delta L^*$), chroma difference ($\Delta C^*$) and hue angle difference ($\Delta h^\circ$) in turquoise (classified in six hue groups).

**Table 5.** The colour difference ($\Delta E_{00}$) of turquoise based on water immersion experiment.

| | | B11 | B18 | G01 | G04 | G08 | G07 | G12 | G14 |
|---|---|---|---|---|---|---|---|---|---|
| original hue angle | | 223.60 | 215.99 | 191.42 | 180.98 | 169.39 | 158.64 | 140.79 | 121.32 |
| original sample images | | | | | | | | | |
| original colour parameters | $L^*$ | 77.01 | 83.58 | 68.35 | 65.06 | 62.25 | 55.86 | 66.57 | 70.03 |
| | $a^*$ | −26.23 | −22.99 | −29.06 | −26.77 | −22.05 | −22.93 | −17.33 | −16.31 |
| | $b^*$ | −24.98 | −16.69 | −5.587 | −0.16 | 4.13 | 8.97 | 14.14 | 26.80 |
| $\Delta E_{00}$ | | 9.20 | 7.51 | 0.82 | 2.48 | 1.72 | 1.82 | 0.61 | 0.53 |
| SG | | 2.69 | 2.63 | 2.87 | 2.81 | 2.81 | 2.72 | 2.78 | 2.79 |
| colour simulation of turquoise after the experiment | | | | | | | | | |

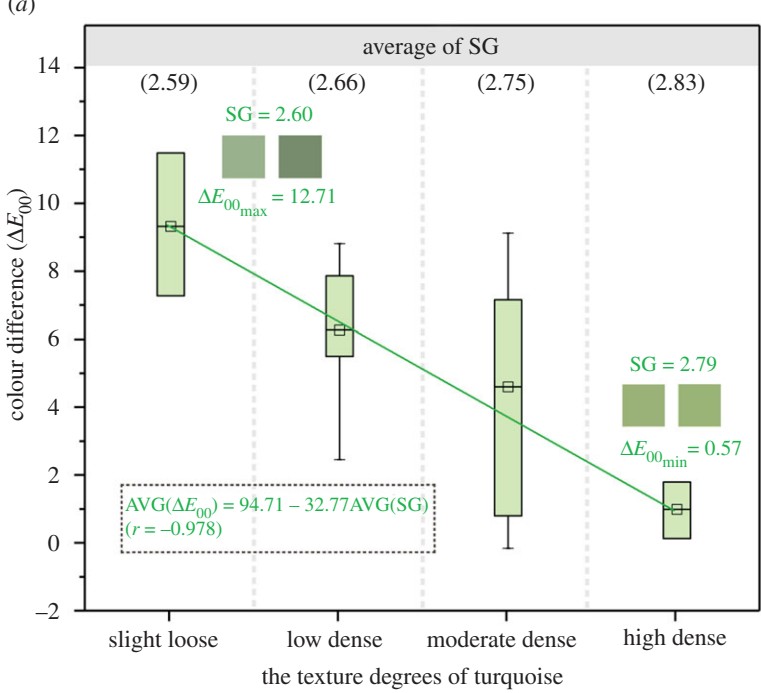

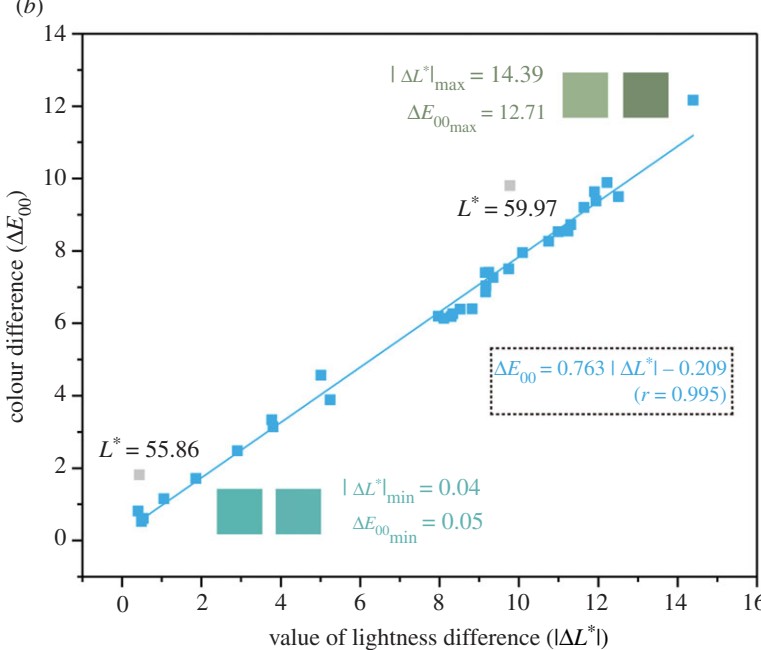

**Figure 11.** (a) Box chart of colour difference ($\Delta E_{00}$) with four texture degrees; the AVG of $\Delta E_{00}$ for each texture degree is negatively correlated with the AVG of SG. (b) The correlation between the value of lightness difference $|\Delta L^*|$ and colour difference $\Delta E_{00}$, where the discrete grey points are due to a lower lightness ($L^* < 60$).

### 3.4.3. Colour difference analysis

Colour difference is a comprehensive description of differences between a control colour and a sample colour, involving lightness differences, chroma differences and hue angle differences. The CIE DE2000 colour difference ($\Delta E_{00}$) is well correlated with the visual judgement, and we can easily distinguish the difference between two kinds of colours with a large $\Delta E_{00}$. The colour difference ($\Delta E_{00}$) is calculated using the CIE DE2000 formula (table 5), indicating that the colour of blue turquoise varies greatly with increasing absorbed-water content; this variation is easily perceived by naked-eye observation due to a large $\Delta E_{00}$ of 9.20. The SGs of all samples are divided into three grades in accordance with the Turquoise Grading of the Chinese National Standard [61]. Specifically, SG > 2.70

indicates an extremely dense texture, $2.50 < SG \leq 2.70$, a dense texture and $SG \leq 2.50$, a loose texture. Therefore, all turquoise samples in this paper were divided into four degrees; e.g. high-dense ($SG > 2.80$), moderate-dense ($2.70 < SG \leq 2.80$), low-dense ($2.60 < SG \leq 2.70$) and slight-loose ($2.50 < SG \leq 2.60$).

The box chart in figure 11$a$ indicates that moderate- and low-dense turquoises exhibit great variation in colour difference with a large range; this may be attributed to the uneven absorbed-water content in the original turquoise. The average SG is negatively correlated with the average $\Delta E_{00}$ in each degree, which can be expressed as (figure 11$a$)

$$\overline{\Delta E_{00}} = 94.71 - 32.77\overline{SG} \quad (r = -0.987). \tag{3.4}$$

The result indicates that water-filled pores are well developed in slight-loose turquoise with a low SG, giving rise to obvious colour variation with increasing absorbed-water content. Meanwhile, the colour difference ($\Delta E_{00}$) follows a positive trend with the value of lightness difference ($|\Delta L^*|$) for all turquoise samples (figure 11$b$)

$$\Delta E_{00} = 0.763|\Delta L^*| - 0.209 \quad (r = 0.995). \tag{3.5}$$

The $r$ value is high at 0.995, indicating that lightness difference can exert a significant impact on colour difference caused by water immersion experiment. The lightness of the two samples, deviating from the fitting line (shown as grey points in figure 11$b$), displays a weaker influence on colour difference due to their darker lightness ($L^* < 60$). In summary, loose turquoise with a lower SG tends to possess a greater colour difference with a significant decrease in lightness. For blue turquoise, an increasing amount of absorbed water results in a decrease in lightness (with less white) and an increase in chroma (a deeper blue). Since high lightness has an adverse impact on colour blue and results in a low-level quality of turquoise's blue, the result also means that increasing absorbed-water content can enhance colour blue quality of turquoise, which helps it to show fancy and brilliant blue.

# 4. Conclusion

In order to establish a suitable evaluation system in the future research, with an emphasis on colour quality of turquoise, it is necessary to better understand the causes of colour transition in turquoise. This study puts forward an effective and non-destructive method to quantitatively analyse the colour of turquoise, and combines with chemical and spectral analyses, determining the impact of trace metal cations and absorbed water on colour of turquoise. Our conclusions are as follows:

(1) Fe content correlates well with Al content in green-hue turquoise ($R^2 = 0.980$), indicating that substitution of Al by $Fe^{3+}$ results in the solid-solution series of turquoise-chalcosiderite. With increasing Fe content, the double absorption peaks at 422 and 428 nm merge into a strong narrow band at 426 nm in UV–Vis spectra. The band at 426 nm shifts into 428 nm with increasing V content, corresponding to a colour transition from blue to green in turquoise. $Zn^{2+}$ suppresses the blue in turquoise, $V^{3+}$ enhances absorption in the violet, blue and orange regions of the spectra, and $Cr^{3+}$ enhances absorption in the red and green regions of the spectra, all of which result in the vivid greenish yellow of faustite.

(2) Substitutions of Al by medium-sized trivalent cations (primarily $Fe^{3+}$ and $V^{3+}$) enhance the polarity of the phosphate group ($(PO_4)^{3-}$), increasing absorption strength in the infrared spectra for analogues of turquoise. FTIR spectra provide a highly efficient way to distinguish faustite ($R_{OH} > 1.000$) from the turquoise group using the reflectivity ratio ($R_{OH}$) of the double absorption peaks at 781 and 833 $cm^{-1}$.

(3) Increasing absorbed-water content has a negative impact on the lightness of all turquoise samples, whereas, it enhances chroma for blue turquoise. Loose turquoise with a low SG tends to possess a large colour difference, especially showing a significant decrease in lightness. Increasing absorbed-water content can enhance colour blue quality of turquoise, which helps it to show fancy and brilliant blue.

Ethics. This article does not present research with ethical considerations.

Data accessibility. Data available from the Dryad Digital Repository: (https://doi.org/10.5061/dryad.qjq2bvqd5) [62].

Authors' contributions. X.W. totally conceived and designed the study, carried out experiments, performed data analysis and drafted the manuscript. Y.G. provided valuable suggestions throughout, critically revised the manuscript and supported on samples. All authors gave final approval for publication.

Competing interests. The authors declare no competing interests.

Funding. There is no funding to report for the submission.

Acknowledgements. We gratefully acknowledge the patient guideline and rigorous suggestions from Y. Guo and support from the Lab of Gemological Research at School of Gemmology, China University of Geosciences, Beijing. We also gratefully acknowledge for valuable reference on the crystal structure of turquoise from A. Y. Abdu.

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
