## [Peer Review File · Royal Society Open Science]

Review History

RSOS-201110.R0 (Original submission)

Review form: Reviewer 1

Is the manuscript scientifically sound in its present form?

Yes

Are the interpretations and conclusions justified by the results?

Yes

Is the language acceptable?

Yes

Do you have any ethical concerns with this paper?

No

Have you any concerns about statistical analyses in this paper?

No

Recommendation?

Major revision is needed (please make suggestions in comments)

Comments to the Author(s)

Comments on title, abstract and references.

The title of the article is informative and relevant.

The purpose of the article is clear, as well as what was found and how they did it.

The references used are relevant, recent, not all correctly referenced and appropriate for the key included studies.

Reference 7 is incorrect.

In some of the references it is necessary to include the issue number because this information was not included.

Comments on introduction

The authors clearly describe what is already known about this topic, the research question is clearly highlighted, and the research question is adequately justified according to what is already known about the topic.

Comments on methodology

The subject selection process is clear, the variables are defined and measured appropriately, the study methods are valid and reliable, and there is enough information to replicate the study. Authors should use the equation editor properly to edit equation 3.

Comments on results and discussion

The data is presented in an appropriate way, the text added to the data is not repetitive, and the results are clearly described from a practical point of view.

The results are discussed from multiple angles without being over-interpreted.

The technical variables of the addresses *a*, *b* and *c* must be written using italic fonts.

The phrase "three-dimension" has that be changed to "three-dimensions".

"d-d electron" must use italic fonts for "d" letters.

"r" must use italic fonts.

Comments on conclusions

The conclusions respond to the objectives of the study and are supported by the results achieved. Unfortunately, the authors do not report on possible future research areas for this publication.

Review form: Reviewer 2

Is the manuscript scientifically sound in its present form?

Yes

Are the interpretations and conclusions justified by the results?

Yes

Is the language acceptable?

No

Do you have any ethical concerns with this paper?

No

Have you any concerns about statistical analyses in this paper?

Yes

Recommendation?

Major revision is needed (please make suggestions in comments)

Comments to the Author(s)

This manuscript presents the result of a set of investigations that relate variation in the chemical composition (based on certain trivalent ions and water content) and the appearance of color (quantified by certain parameters) in different types of gem-quality turquoise/ jade stone samples. The investigations rely on experimental data from techniques such as EDXRF, UV-Vis, IR, SEM and regression analyses from the same data. Though the manuscript presents many details, it lacks certain clarity that is required for a scientific article for publication in a journal meant for general chemistry readers, rather it reads like an internal technical report. Therefore, I could not recommend it in the current version but recommend a major revision by the authors considering comments below. 1) There is insufficient literature presented that helps the readers to understand what is already known in the area, the need for the current work and to appreciate the conclusion drawn in such a scientific study. 2) Method sections require additional details of the sample states (sample preparation, solid powder or single block etc. for UV-Vis), regression analysis description etc. A separate section with appropriate references in the SI will be helpful that give a full description of all the parameters such as lightness, color CIE coordinates of different versions, etc...). Most importantly, the physical significance of each parameter has to be elaborated in order to understand their role in the color appearance and to appreciate the results and discussion part. 3) All the spectroscopic peak assignments require appropriate references and may be presented in a table form. 4) The manuscript has many typos and grammatical errors, which require special attention.

Decision letter (RSOS-201110.R0)

Dear Miss Wang:

Title: The Impact of Trace Metal Cations and Absorbed Water on Color Transition of Turquoise
Manuscript ID: RSOS-201110

The editor assigned to your manuscript has now received comments from reviewers. I apologise that this has taken longer than usual. We would like you to revise your paper in accordance with the referee and Subject Editor suggestions which can be found below (not including confidential reports to the Editor). Please note this decision does not guarantee eventual acceptance.

Please submit your revised paper before 25-Nov-2020. Please note that the revision deadline will expire at 00.00am on this date. If we do not hear from you within this time then it will be assumed that the paper has been withdrawn. In exceptional circumstances, extensions may be possible if agreed with the Editorial Office in advance. We do not allow multiple rounds of revision so we urge you to make every effort to fully address all of the comments at this stage. If deemed necessary by the Editors, your manuscript will be sent back to one or more of the original reviewers for assessment. If the original reviewers are not available we may invite new reviewers.

On behalf of the Subject Editor Professor Anthony Stace and the Associate Editor Dr Annette Trunschke.

RSC Associate Editor:
Comments to the Author:
(There are no comments.)

RSC Subject Editor:
Comments to the Author:
(There are no comments.)

Reviewers' Comments to Author:
Reviewer: 1
Comments to the Author(s)
Comments on title, abstract and references.

The title of the article is informative and relevant.

The purpose of the article is clear, as well as what was found and how they did it.

The references used are relevant, recent, not all correctly referenced and appropriate for the key included studies.

Reference 7 is incorrect.

In some of the references it is necessary to include the issue number because this information was not included.

Comments on introduction

The authors clearly describe what is already known about this topic, the research question is clearly highlighted, and the research question is adequately justified according to what is already known about the topic.

Comments on methodology

The subject selection process is clear, the variables are defined and measured appropriately, the study methods are valid and reliable, and there is enough information to replicate the study. Authors should use the equation editor properly to edit equation 3.

Comments on results and discussion

The data is presented in an appropriate way, the text added to the data is not repetitive, and the results are clearly described from a practical point of view.

The results are discussed from multiple angles without being over-interpreted.

The technical variables of the addresses a, b and c must be written using italic fonts.

The phrase "three-dimension" has that be changed to "three-dimensions".

"d-d electron" must use italic fonts for "d" letters.

"r" must use italic fonts.

Comments on conclusions

The conclusions respond to the objectives of the study and are supported by the results achieved. Unfortunately, the authors do not report on possible future research areas for this publication.

Reviewer: 2

Comments to the Author(s)

This manuscript present the result of a set of investigations that relate variation in the chemical composition (based on certain trivalentions and water content) and the appearance of color (quantified by certain parameters) in different types of gem-quality turquoise/ jade stone samples. The investigations rely on experimental data from techniques suchas EDXRF, UV-Vis, IR, SEM and regression analyses from the same data. Though the manuscript presents many details, it lacks certain clarity that is required for a scientific article for publication in a journal meant for general chemistry readers, rather it reads like an internal technical report. Therefore, I could not recommend it in the current version but recommend a major revision by the authors considering comments below. 1) There is insufficient literature presented that help the readers to understand what is already known in the area, the need for the current work and to appreciate the conclusion drawn in such a scientific study. 2) Method sections require additional details of the

sample states (sample preparation, solid powder or or single block etc. for UV-Vis), regression analysis description etc. A separate section with appropriate references in the SI will be helpful that give a full description of all the parameters such as lightness, color CIE coordinates of different versions, etc...). Most importantly, the physical significance of each parameter has to be elaborated in order to understand their role in the color appearance and to appreciate the results and discussion part. 3) All the spectroscopic peak assignments require appropriate references and may be presented in a table form. 4) The manuscript has many typos and grammatical errors, which require special attention.

Author's Response to Decision Letter for (RSOS-201110.R0)

See Appendix A.

Decision letter (RSOS-201110.R1)

This year has been very difficult for everyone, and we want to take the opportunity to thank you for your continued support in 2020.

The Royal Society Open Science editorial office will be closed from the evening of Friday 18 December 2020 until Monday 4 January 2021. We will not be responding during this time. If you have received a deadline within this time period, please contact us as soon as possible to allow us to extend the deadline. If you receive any automated messages during this time asking you to meet a deadline, we offer apologies and invite you to respond after the festive period or during normal working hours.

With our best for a peaceful festive period and New Year, and we look forward to working with you in 2021.

Dear Miss Wang:

Title: The Impact of Trace Metal Cations and Absorbed Water on Color Transition of Turquoise
Manuscript ID: RSOS-201110.R1

It is a pleasure to accept your manuscript in its current form for publication in Royal Society Open Science. The chemistry content of Royal Society Open Science is published in collaboration with the Royal Society of Chemistry.

Royal Society of Chemistry
Thomas Graham House

Science Park, Milton Road
Cambridge, CB4 0WF
Royal Society Open Science - Chemistry Editorial Office

On behalf of the Subject Editor Professor Anthony Stace and the Associate Editor Dr Annette Trunschke.

RSC Associate Editor
Comments to the Author:
(There are no comments.)

Reviewer(s)' Comments to Author:

Appendix A

Dear editors,

It's our pleasure to receive your e-mail about the comments concerning our manuscript entitled **“The impact of trace metal cations and absorbed water on color transition of turquoise (RSOS-201110).**

We are grateful to all editors for their suggestions and comments on the manuscript. Please convey our best regards to the reviewers. They have put forward many valuable and informative suggestions for us to advance the manuscript and further research. During revising this manuscript, all the comments have been taken into consideration and have been carefully answered in the manuscript. Thanks for your sincere work on our manuscript again. If you have any questions, please feel free to contact us, we will do our best to offer you a reasonable solution. At last, I sincerely hope that the revised manuscript will be accepted in *Royal Society Open Sciences*. The details about the modifications are shown in the revised manuscript (corrections have been marked in red color), and the answers to the reviewer's questions are shown as follows.

Looking forward to hearing from you.

Best regards!

Sincerely yours,

Xueding Wang

20009180027@cugb.edu.cn

Ying Guo

guoying@cugb.edu.cn

China University of Geosciences Beijing

Response to Reviewer #1 and Reviewer #2 Comments:

Response to Reviewer #1:

[1] Comments on the title, abstract and references

“The references used are relevant, recent, not all correctly reference and appropriate for the key included studies.”

Response: Thanks for the comments. You suggest that there are several references, not all correctly cited and appropriate for the key. We guess that the mentioned references are almost about ED-XRF. Firstly we pray for your understanding that ED-XRF have been widely used in most of fields, while it's still a young technique in gemmology and not frequently used to study the chemical composition of turquoise. As gemmology is a secondary discipline of geology, we usually cite several references in fields of geology and mineralogy to support our study. The less-appropriate references are cited to explain the feasibility and effectiveness of ED-XRF analysis in many sciences (such as geology, materials science, and mineralogy), and so we believe it is also helpful for our study. But if you still suggest that we should delete them, we will respect and adopt your precious comments.

“Reference 7 is incorrect.”

Response: Reference 7 has been corrected as follows:

7. J. Kolitsch and G. Giuseppetti. 2000. The crystal structure of faustite and its copper analogue turquoise. *Mineralogical Magazine*. 64(5), 905-913. (DOI: 10.1180/002646100549733)

Authors

Response to reviewer #1: Thanks again for the careful reading. We've corrected the mistake in reference #7.

“In some of the references, it is necessary to include the issue number because this information was not included.”

Response: Thanks for the comments in detail. We have carefully re-checked the references and deleted fourteen papers which don't have certain issue numbers. The cited orders of several references have been adjusted in order to make the introduction more logical, and we have newly added twenty-four articles which are more appropriate for our study. All the revision on references are shown in the annex table (in the end of the letter). Besides that, we have modified the necessary information (primarily the issue number) for all the references, as you suggested. However, reference 61 in the revised manuscript is a Chinese standard document about turquoise grading, and we are sorry that it only has a standard number (GB/T 36169-2018), without an issue number. This standard document is a good guideline for us to classify turquoise texture quality using its specific gravity, so we sincerely pray for keeping the cited.

[2] Comments on the methodology

“Authors should use the equation editor properly to edit equation 3.”

Response: Thanks for the comment in detail. We've re-edited equation 3 in a proper way and corrected the mistakes in it. The revised equation 3 is as follows:

$$\Delta E_{00} = \sqrt{\left(\frac{\Delta L^*}{K_L S_L}\right)^2 + \left(\frac{\Delta C^*}{K_C S_C}\right)^2 + \left(\frac{\Delta H^*}{K_H S_H}\right)^2} + R_T \left(\frac{\Delta C^*}{K_C S_C}\right)^2 \left(\frac{\Delta H^*}{K_H S_H}\right) \quad (3)$$

[3] Comments on results and discussion

"The technical variables of the address a,b and c must be written using italic fonts."

"The phrase 'three-dimension' has that be changed to 'three-dimensions'. 'd-d electron' must use italic fonts for 'd' letters. 'r' must use italic fonts."

Response: Thanks for your careful reading and the comments on writing. We've all re-edited them in required italic fonts.

[4] Comments on conclusion

"Unfortunately, the authors do not report on possible future research areas for the publication."

Response: Thanks for the valuable comments so much. To be honest with you, these comments well encourage us to develop our research further, not only to complete the objectives or achieving the results, but also to study color of gemstones in a more advanced method. A simple view of possible future research is added on page 9 (in 1st-5th lines), which is described as: "In order to establish a suitable evaluation system in the future research, with an emphasis on color quality of turquoise, it is necessary to better understand the causes of color transition in turquoise. This study puts forward an effective and non-destructive method to quantitatively analyze the color of turquoise, and combines with chemical and spectral analyses, determining the impact of trace metal cations and absorbed water on color of turquoise."

Meanwhile, we prepare a full description of possible future research for you, helping you to understand value of this current study:

(1) Firstly, this study provides an effective and non-destructed method to quantitatively analyze colors of gemstones, which can also expand to study minerals and rocks with complex chemical compositions. Based on the conclusion, we can better understand a color transition from blue to yellow in turquoise, affected by trace metal cations. And it is a basis that we can further study color grading and quality evaluation of turquoise, the latest highlight question in gemmology.

(2) Secondly, the conclusion from FTIR spectra is a valuable and advanced discovery, and provides a high-efficient way to allow faustite to be distinguished from the turquoise group, using absorption characteristics due to the OH bending vibration. Besides that, we can further study the correlation between color and FTIR spectra in turquoise, and make an attempt to determine color quality of turquoise through FTIR spectra.

(3) Thirdly, it is necessary to take research on absorbed water in turquoise since it is closely associated with color-quality maintenance. The conclusion confirms that blue turquoise displays more brilliant color due to a increase in absorbed-water content. On the contrary, it implies that losing absorbed water may have an adverse impact on color in turquoise. Meanwhile, a fact is that

the color and appearance of turquoise are easily affected or even destroyed when air humidity declines seriously. Therefore, we put forward a practical point that turquoise should be stored and exhibited in a gentle environment, with constant temperature and humidity, as to stop turquoise losing its absorbed water in a too dry air. Based on the conclusion in water immersion experiment, we would further study the impact of air humidity on turquoise's color, explaining how to protect appearance of turquoise.

(4) In addition, the conclusion implies that we should take into consideration about a standard environment, where we operate color grading for an unknown-grade turquoise there. Air humidity as well as color temperature of illumination may have a impact on absorbed-water content in turquoise, and these external factors should be concentrated on in the future research.

In summary, it is significant that we should completely understand the theory and knowledge, including color causing and color transition of turquoise. Based on all of these, we hope to establish a valuable evaluation system with an emphasis in turquoise's color quality. Not only is it a latest and practical topic in gemmology, but also is a development from gemmology science to gemmological technology.

Response to Reviewer #2:

[1] Comments on references

“There is insufficient literature presented that helps the readers to understand what is already known in the area, the need for the current work and to appreciate the conclusion drawn in such a scientific study.”

Response: Thanks for the comments on literature. We have re-checked all of the references, and deleted fourteen papers due to their unclear issue numbers. Considering the insufficient literature, we have currently added twenty-four scientific articles in the revised manuscript, which better give assistance to our study. All the revision of references are presented in the annex table (in the end of the letter). Here is an explanation of the added references in the revised manuscript:

- 1) Reference [1] on page 1 is newly added to introduce the important value of turquoise in China. The original reference [1] is shifted to the cited [2] in the revised manuscript.
- 2) Reference [5] on page 1 is newly added to introduce an important origin (Iran) of turquoise, which we didn't mention in the original manuscript.
- 3) Reference [10] on page 1 is newly added to better introduce the current research on color causing of turquoise. The original reference [10] has been deleted as it lacks necessary information.
- 4) Reference [19] on page 2 is newly added to introduce an important technique (LA-ICP-AES) to study the chemical composition of turquoise. The sentence at the position is revised to: “**Laser ablation inductively coupled emission spectrometer (LA-ICP-AES) is used to study the geochemical characteristics of trace elements and rare-earth elements in modern turquoise, to determine its origin**”. The originally cited [19] has been deleted as it lacks necessary information.
- 5) Reference [22] on page 2 is newly added to explain the feasibility of ED-XRF analysis in field of materials science. The originally cited [22] has been deleted.

6) Reference [24] on page 2 is newly added to explain the feasibility of ED-XRF analysis in field of environmental science. The originally cited [24] has been deleted.

7) Reference [26] on page 2 is newly added to explain current research on the chemical composition of turquoise using ED-XRF, which also gives assistance to the results and discussion of ED-XRF (on page 4).

8) References [30-32] on page 2 are newly added to explain what are advantages about FTIR, and why we choose to use it in this study. The added sentences at the position are described to: "Fourier-transform infrared (FTIR) spectroscopy is a valuable technique in studying hydrated minerals, and is sensitive to the hydroxyl (OH) group; this type of spectroscopy easily distinguishes OH with H₂O molecules in the structure. Moreover, FTIR is well used to determine the origin of natural turquoise". These references are also beneficial to the results and discussion of FTIR (on page 6), well helping to determine the spectral peak assignments. The cited [5] in the original manuscript has been shifted to the cited [32] in the revised one)

9) References [35-40] on page 2 are newly added to introduce a wide utilization of the CIE color system we used in the study. The sentence there is revised to: "The CIE 1976 L*a*b* uniform color system is the most popular system for color measurement and analysis, recommended by the International Commission on Illumination (CIE)". Taking into consideration that the journal is meant for general chemistry readers, as you reminded, we believe that these references may help the readers to understand the basis knowledge and theory of the color system. And they are efficiently used to explain the physical significance of color parameters (e.g., lightness, chroma, colorimetric coordinates, etc.) mentioned in the method part (on page 3).

10) References [47,48] on page 2 are newly added to explain the feasibility of the CIE color system in color research on amethyst and diamond. The sentence there is revised to: "The color system is widely applied in the study of gem color, including jadeite-jade [41-43], peridot [44,45], tourmaline [46], amethyst [47] and diamond [48]". Due to wide feasibility of the color system in gem's color research, we believe that it is similarly sensible to our study on turquoise color.

11) References [49,50] on page 2 are newly added to provide a description of the color difference formula, CIE DE2000, and to introduce its efficient utilization in research of soil and gem. The sentence there is revised to: "The formula CIE DE2000 can be used to express color difference quantitatively and effectively [49-51]". Comparing with CIE LAB (the other common formula) mentioned in the method part (on page 3), these references help readers to understand the advanced feature of CIE DE2000.

12) Reference [53] on page 6 (in 2nd line) is newly added to introduce a stepwise regression analysis, which is used in the results and discussion of UV-Vis spectra. The sentence there is described as: "A stepwise regression method is often used to screen out variables with significant correlation and to judge the contributions of independent variables to dependent variables [53]".

13) Reference [54] on page 6 (in 20th line) is newly added to give assistance to the infrared spectral analysis. Since there are some special peaks in the infrared spectra of turquoise, which seem like stretching vibrations caused by the CO₃²⁻ unit, this reference provides the evidence to our study. The sentence at the position is revised to: "The C-O stretching vibrations of the carbonate group (CO₃)²⁻

are located near 1450 cm^{-1} , with weak absorption, which can possibly be attributed to the presence of the admixed carbonate on the chalcosiderite-faustite surface [54].”

14) Reference [57] on page 6 (in 38th line) is newly added to introduce the feasibility of K-means clustering analysis in field of geology, which is good at classifying object’s characteristics quickly.

15) Reference [58] on page 6 (in 40th line) is newly added to introduce the feasibility of Fisher discriminant analysis in gemmology, which is good at verifying a accuracy of classification.

16) References [59,60] on page 7 (in 21th line) are newly added to introduce the current work of SEM in the texture study on turquoise, which contribute to the results and discussion of water immersion experiment.

[2] Comments on important details

“Method sections require additional details of the sample states (sample preparation, solid powder or single block etc. for UV-Vis), regression analysis description etc. A separate section with appropriate references in the SI will be helpful that give a full description of all the parameters such as lightness, color CIE coordinates of different versions, etc. Most importantly, the physical significance of each parameter has to be elaborated in order to understand their role in the color appearance and to appreciate the results and discussion part.”

Response: Thanks for the comments. We’re grateful that you remind us of many required and significant details which are important to our manuscript. We have completed all the answers you suggested, and revised in the manuscript. Here is a point-by-point description as follows:

Comment: “...require additional details of the sample states (sample preparation, solid powder or single block etc. for UV-Vis), regression analysis description etc. ...”

Response: Actually, all samples were prepared as single block, and well polished with a smooth, clear surface, which is beneficial to the chemical and spectral measurement. Considering the requirement of colorimetric analysis, each sample possesses a tested area with a diameter of 6 mm.

The additional information of the sample state (on page 2 and 3) is in red shading, with annotations on the right in the revised manuscript.

1) The sentences on page 2 and 3 are revised to: “Fifteen samples of turquoise had a round-cabochon shape, while the others had a round-plate shape. All were well polished, with a smooth, clear surface, conducive to chemical and spectral investigation.”

2) The sample state for ED-XRF on page 3 is described as: “Each sample was prepared as single block, whose test area was a circle with a diameter of 3 mm.”

3) The sample state for UV-Vis on page 3 is described as: “Each sample was prepared as single block, with a polished, smooth surface.”

4) The sample state for IR on page 3 is described as: “The sample states were the same as above.”

5) The added description of the sample state for colorimetric analysis is revised to: “The tested area of a single sample was a circle with a diameter of 6 mm.”

The description of regression analysis helps readers to understand its role in the result of UV-Vis analysis. The sentences on page 6 (in 1st–5th lines) are revised to: “A stepwise regression method is often used to screen out variables with significant correlation and to judge the contributions of independent variables to dependent variables [53]. Therefore, we used this method to study the contributions of color-causing metal elements (Cu, Fe, Zn, V, and Cr) to b^* (the colorimetric coordinate) in turquoise and to explain which element more sensitively influences the hue variation, from yellow to blue in turquoise.”

The description of partial correlation analysis helps readers to understand its roles in the result of EDXRF analysis. The sentences on page 5 (in 8th–9th lines) are revised to: “Partial correlation analysis is effective in analyzing the correlations among different variables when taking control of a relative variable.”

Comment: A separate section with appropriate references in the SI will be helpful that give a full description of all the parameters such as lightness, color CIE coordinates of different versions, etc. Most importantly, the physical significance of each parameter has to be elaborated in order to understand their role in the color appearance and to appreciate the results and discussion part.”

Response: Thanks for the comments in detail. We are sorry that our original manuscript lacks certain clarity as you suggested. Thinking carefully of your precious comments, we would like to provide a detailed explanation of the color system at first.

The CIE 1976 $L^*a^*b^*$ uniform color system is a standard for the quantitative expression of any color using three color parameters (L^* , a^* , and b^*), currently recommended by the International Commission on Illumination (CIE). The color system is developed from the CIE 1931 XYZ color system, and its parameters are calculated using XYZ tristimulus values, establishing a conversion between color parameters with wavelength of visual light. At present, the color system is commonly applied in many field of research (e.g., computer, printing, soil, and gem), giving assistance to quantitative color analysis.

It comprises a three-dimensional spherical color space, with colorimetric coordinates (a^* and b^*) in the horizontal direction and lightness (L^*) in the vertical direction. We added a figure to illustrate what the color system looks like, helping you to understand this part well (see **Fig.R1(a)**). If it is necessary, we are willing to add it into the article in the future.

Fig.R1 The color system

Here is a full description of parameters and their physical significance:

- a. Lightness represents color variation between darkest black ($L^*=0$) and lightest white ($L^*=100$), and a growth in lightness value means that the color changes brighter (Y. N. Vodyanitskii 2016).
- b. The colorimetric coordinate a^* describes a color variation from red ($+a^*$) to green ($-a^*$), while the colorimetric coordinate b^* describes a color variation from yellow ($+b^*$) to blue ($-b^*$). The phase “chromatic diagram” on page 7 is a two-dimensional plane composed of the colorimetric coordinates a^* and b^* . It shows that chroma and hue angle are calculated using coordinates a^* and b^* (see **Fig.R1(b)**). The center in the a^*-b^* plane represents a neutral color, with 50% white and 50% black (M. R. Pointer 1981).
- c. Chroma (C^*) represents saturation variation of single color (e.g., blue), between lightest blue ($C^*=0$) and deepest blue ($C^*=100$). A color with high chroma is more brilliant and intensive (M. Madhura P 2020).
- d. Hue angle (h°) varies from 0 to 2π ($0^\circ-360^\circ$), representing a series of continuous color variation from red, yellow, green, blue to purple. The phrase “5 centers of h° ” on page 4 is a classification method of hue angle. The whole range ($0^\circ-360^\circ$) is divided into 12 sections by single 30° interval, and 12 centers of hue angle in 12 intervals are $15^\circ, 45^\circ, 75^\circ, 105^\circ, 135^\circ, 165^\circ, 195^\circ, 225^\circ, 255^\circ, 285^\circ, 315^\circ,$ and 345° (Jin Yang 2013).
- e. Color difference is a comprehensive description of color change in turquoise, involving lightness differences, chroma differences and hue angle differences. The CIE DE2000 color difference (ΔE_{00}) is well correlated with the visual judgment. We can more easily distinguish difference between two kinds of colors, with a large ΔE_{00} . There is an table to describe the correlation between ΔE_{00} and visual judgement (Xiaowei Zhang 2012).

visual judgment	ΔE_{00}
trace	0~0.5
slight	0.5~1.5
noticeable	1.5~3.0
Very noticeable	3.0~6.0
much	6.0~12.0
Very much	>12

We tried our best to re-edit all of description as you required, and finally added a separate paragraph on page 3 in the revised manuscript:

“The color system comprises a three-dimensional spherical color space, with colorimetric coordinates (a^* and b^*) in the horizontal direction and lightness (L^*) in the vertical direction [35]. Lightness represents color variation between darkest black ($L^*=0$) and lightest white ($L^*=100$), and a growth in lightness value means that the color becomes brighter [38]. The colorimetric coordinate a^* describes a color variation from red ($+a^*$) to green ($-a^*$), while the colorimetric coordinate b^* describes a color variation from yellow ($+b^*$) to blue ($-b^*$) [36]. Chroma (C^*) represents saturation variation of single color (e.g., blue), between lightest blue ($C^*=0$) and deepest blue ($C^*=100$). A color with high chroma is more brilliant and intensive [38]. Hue angle (h°) varies from 0 to 2π ,

representing a series of continuous color variation from red, yellow, green, blue to purple [36]. All the color parameters are psycho-physical parameters without unit.”

[3] Comments on results and discussion

“All the spectroscopic peak assignments require appropriate references and may be presented in a table form.”

Response: Thanks for your important comments. The suggestion we acquire from this comment is that we should add references in the result and discussion of FTIR and UV-Vis spectra, to make sure that all the spectroscopic peak assignments are right, and we also should provide a table, including infrared peaks of the studied turquoise samples. Since all the spectral peaks of UV-Vis spectra are clearly assigned in Fig.4 and Fig.5, we just design a table to present the infrared peaks assignments of the studied turquoise samples, which is added as the third table in the revised manuscript.

Table. 3 Infrared peaks of turquoise samples.

	Turquoise		Chalcosiderite		Faustite	
	B11	B16	G01	G12	G09	G14
H-O-H bending vibration	1637	1637	-	-	-	-
C-O stretching vibration	-	-	1448	1448	1450	1450
	1199	1196	-	-	-	-
P-O stretching vibration	1121	1123	1118	1117	1119	1121
	1059	1059	1051	1057	1061/1043	1061/1043
	1007	1007	1009	1009	1005	1003
O-H bending vibration	833	833	833	833	831	798
	781	781	781	781	796	
	650	650	646	-	648	646
	608	608	-	-	623	623
P-O stretching vibration	571	569	565	567	588	590
	536	534	-	536	536	553
	484	480	478	480	480	480
	449	451	449	451	453	453

All the spectroscopic peak assignments are followed by a reference in the revised manuscript (shown in red shading), and here are some examples:

1) The sentence on page 5 (in 20th line) is revised to: “For Cu-rich turquoise [Fig. 4(a)], the absorption band in the orange-red region occurs around 673 nm due to the *d-d* electron transition of Cu²⁺ [6].”

2) The sentence on page 5 (in 24th line) is revised to: “In addition, the double absorption peaks at 422 and 428 nm in the violet-blue region are caused by the electron transition of Fe³⁺ (⁶A₁→⁴E and

${}^4A_1({}^4G)$, while the weak absorption band in the ultraviolet region of 370 nm is caused by the Fe^{3+} electron transition (${}^6A_1 \rightarrow {}^4E({}^4D)$) and charge transferring from O^{2-} to Fe^{3+} [17].”

3) The sentence on page 5 (in 30th line) is revised to: “The weak band in the blue region at 470–480 nm is due to the special Fe^{3+} lattice position [6].”

4) The sentence on page 5 (in 41th line) is revised to: “The electron transition of V^{3+} produces two absorption bands in the orange (620 nm) and violet-blue regions (420–460 nm) [13].”

5) The sentence on page 5 (in 45th line) is revised to: “Small levels of Cr in faustite can generate a narrow absorption band in the green region of 568 nm, and work with Cu to produce a broad absorption band in the red region of 683 nm [14].”

[4] Comments on the writing errors

“The manuscript has many typos and grammatical errors, which require special attention.”

Response: Thanks for your patient comments on typo and grammar. We are so sorry for the non-professional translation in the original manuscript. In order to improve its grammar and coherence, we have invited a professional British editor team (The Charlesworth Author Services Team), to re-translate and re-edit our manuscript. As you can see, a large amount of words and phrases are in red, reflecting the revision marks, and some of them are presented in the below table, reflecting a comparison between errors and revision.

Response to reviewer (comments on typos and grammatical errors)

Section	Location	Errors/improper	Revision
Full paper	-	spectrum	spectra
	-	various color	various colors
Institution	page 1 (below writers)	Geociences	Geosciences
Summary	page 1 (in 5 th line)	The	Sample
	page 1 (in 6 th line)	performs	exhibits
	page 1 (in 8 th line)	as well as	and
	page 1 (in 9 th line)	cause	are responsible for
	page 1 (in 10 th line)	trivalent cations	medium-sized trivalent cations
		higher	strong
	page 1 (in 11 th line)	strength of (deleted)	in the
		can distinguish faustite	allow faustite to be distinguished
	page 1 (in 13 th line)	with value	with a value
		however	while
		shows	has a
	page 1 (in 14 th line)	Increasing	An increasing amount of
	page 1 (in 15 th line)	lower	a low
page 1 (in 16 th line)	with significant decrease	with a significant decrease	

Note: Since there are many modifications in the revised manuscript, we list several significant revision on typos in the below table, other unlisted modification on improper expression are shown in red in the revised manuscript.

Section	location	Errors	Revision
Introduction	Page 1 (in 20 th line)	associated with	prevalent in
	Page 1 (in 30 th line)	the most popular	the most popular constituents
	page 2 (in 1 st line)	crystal	crystalline
	page 2 (in 2 nd line)	increasing	an upward trend in
	page 2 (in 7 th line)	alumina	variscite
	page 2 (in 9 th –10 th line)	multi-receiver plasma mass spectrometer (MC-LCP-MS)	multi-receiver inductively coupled plasma mass spectrometer (MC- ICP -MS)
	page 2 (in 11 th –12 th line)	Laser ablation-plasma mass spectrometer (LA-ICP-MS)	Laser ablation inductively coupled emission spectrometer (LA-ICP- AES)
	page 2 (in 13 th line)	Electron probe	Electron microprobe analysis
Materials and methods	Page 3 (in 17 th line)	Fourier Infrared	Fourier- transform infrared
	page 4 (in 1 st line)	environment	experimental environment
Results and discussion	page 4 (in 12 th line)	the oxide	turquoise oxide
	page 4 (in 16 th line)	with average of 20.008%	with an average of 20.008% (several similar modifications are not mentioned again)
	page 4 (in 22 th line)	distort XO6	distorted XO6
	page 4 (in 31 th line)	three-dimension	three- dimensions
	page 4 (in 33 th line)	12 center hue angles	12 centers of hue angles
	page 4 (in 39 th line)	varies slight and reaches	varies slightly , reaching
	page 4 (in 40 th line)	Fe is rich	There exists a high Fe content
	page 4 (in 41 th line)	As a rare element, Zn is only rich in faustite	Because Zn is a rare element, only faustite is rich in Zn
	page 5 (in 3 rd line)	showing	reflecting
	page 5 (in 5 th line)	Fe ³⁺ to substitute Al	substitution of Al by Fe³⁺
	page 5 (in 7 th line)	influence the content of Al	result in a decreasing Al content
	page 5 (in 14 th line)	w(Fe ₂ O ₃)/w(Al ₂ O ₃) value	value of w(Fe₂O₃)/w(Al₂O₃)
	page 5 (in 25 th –26 th lines)	enlarging / enhancing	enlargement / enhancement
	page 5 (in 27 th line)	which displays	reflecting
	page 5 (in 28 th line)	perform	undergo
	page 5 (in 29 th line)	The absorption band nearly 370 nm becomes broader	The absorption band near 370 nm broadens
	page 5 (in 38 th line)	caused by	arising from
	page 5 (in 43 th line)	Minor Cr	Small levels of Cr
	page 5 (in 44 th line)	combine with	work with
page 6 (in 6 th line)	exclude	excluded	
page 6 (in 7 th line)	can well influence	have correlation with	

	page 6 (in 12 th line)	in the range of 2000-400 cm ⁻¹	in 2000-400 cm⁻¹ range
	page 6 (in 16 th line)	1630	1637
	page 6 (in 20 th line)	turquoise	chalcosiderite-faustite
		result in frequency shift and strength change in absorption	cause changes in frequency and absorption strength
	page 6 (in 23 th line)	especially for the phosphate group	especially in the spectral region caused by the phosphate group
	page 6 (in 24 th line)	trivalent cations	small-sized trivalent cations (Al ³⁺)
	page 6 (in 26 th line)	trivalent cations	medium-sized trivalent cations
	page 6 (in 28 th line)	showing	leading to
	page 6 (in 29 th line)	diminishes	reduces
	page 6 (in 30 th line)	lack in peaks at 609 and 648 cm-1, which is an evidence to identify as chalcosiderite	chalcosiderite loses absorption peaks at 609 and 648 cm ⁻¹ , distinguishing from turquoise
	page 6 (in 32 th line)	distinguish	determine
	page 6 (in 37 th line)	is a quick technology to divide	allow the quick division of
	page 6 (in 38 th line)	regularity	common feature
	page 6 (in 40 th line)	accuracy of classification	accuracy of clustering analysis
	page 6 (in 44 th line)	groups	categories
	page 7 (in 8 th line)	reflect	influence
	page 7 (in 10 th line)	-	control color data
		natural	atmospheric
	page 7 (in 11 th line)	taken out	remove
	page 7 (in 12 th line)	-	final color data
	page 7 (in 17 th line)	by hydrostatic weighing	using a hydrostatic weighing scale
	page 7 (in 21 th line)	further analyzing its structural compactness and porosity	especially with respect to its structural compactness and porosity
	page 7 (in 32 th -35 th lines)	grammatical errors in time status	Revised to the past tense.
	page 7 (in 32 th -33 th lines)	All turquoise samples were observed appearance variation	all samples displayed significant color variation
	page 7 (in 43 th line)	chromatic diagram	chromatic diagram (a two-dimensional plane of coordinates a*and b*)
	page 8 (in 24 th line)	referring to	in accordance with
Conclusion	page 8 (in 6 th line)	is well correlated	correlates
	page 9 (in 9 th line)	forming	corresponding to
	page 9 (in 15 th line)	distinguish faustite from turquoise and chalcosiderite with value > 1	distinguish among faustite (R_{0H}>1.000), chalcosiderite and turquoise using the reflectivity ratio (R_{0H})

Annex:

Table. a1 Revision on references

Original reference number	Revised reference number	Added references
1. A. M. Thibodeau 2018	Re-ordered in No.2	1.X. H. Ye. 2014
2. A. M. Thibodeau 2015	Re-ordered in No.3	
3. A. C. Sigleo. 1975	deleted	
4. P. Tallet.2014	No.4	
5. Q. N. Wang. 2018	Re-ordered in No. 32	5.E. M. Gandomani. 2020
6. Q. L. Chen. 2012	No.6	
7. U. Kolitsch. 2000	No.7	
8. Y. A. Abdu. 2011	No.8	
9. Y. A. Abdu. 1998	No.9	
10. L. J. Luan. 2004	deleted	10.Q. Guo. 2014
11. H. F. Zhang. 1982	No.11	
12. H. Q. Xue. 1995	No.12	
13. H. Yang. 2019	No.13	
14. Y. Zhou. 2014	No.14	
15. J. C. Pei. 2019	No.15	
16. J. Liu. 2018	No.16	
17. B. Reddy. 2006	No.17	
18. S. Hull. 2018	No.18	
19. X. He. 2011	deleted	19.Y. X. Heng. 2016
20. X. T. Li. 2019	deleted	
21. L. Chassapis. 2008	No.21	
22. K. Pytlakowska. 2016	deleted	22.M. Nageeb Rashed. 2017
23. S. J. Barnes. 1993	No.23	
24. C. Chou. 2004	deleted	24.T. T. Gan. 2020
25. E. Fritsch. 1999	No.25	
26. J. Q. Yu. 2019	deleted	26.H. Sabbaghi. 2018
27. G. Choudhary. 2010	No.27	
28. L. Liu. 2018	No.28	
29. J. Yan. . 2015	No.29	
30. X. A. Li. 1984	deleted	
31. E. Crespoe-Feo. 2010	Re-ordered in No.20	
32. K. L. Guo. 2010	deleted	
33. Y. Guo. 2017	deleted	33.Q. L. Chen. 2020
34. Y. Guo. 2018	Re-ordered in No.42	34.R. L. Frost. 206
35. J. Tang. 2018	deleted	35.M. R. Pointer. 1981
36. J. Tang. 2019	Re-ordered in No.44	36.N. P. Kirillova. 2014
37. J. Tang. 2019	Re-ordered in No.45	37.Y. N. Vodyanitskii . 2016
38. Y. L. Yang. 2016	Re-ordered in No.46	38.M. Madhura P. 2020

39. J. Y. Han. 2019	deleted	39.S. S. Guan. 1999
40. Z. Y. Sun. 2014	deleted	40.H. S. Xu. 2001
41. K. McLAREN. 2008	Re-ordered in No.49	
42. O. E. Pecho. 2016	Re-ordered in No.51	
43. Y. Jin. 2013	Re-ordered in No.52	43.X. Pan. 2019
44. H. H. Adler. 1963	deleted	
45. W. J. Chen. 2018	Re-ordered in No.30	
46. J. Ren. 2015	Re-ordered in No.31	
47. Y. Guo. 2018	Re-ordered in No.55	47.R. P. Chen. 2020
48. Y. Guo. 2016	Re-ordered in No.56	48.F. K. Liu. 2020
49. Y. Guo. 2016	Re-ordered in No.41	
50. Turquoise Grading	Re-ordered in No.61	50.C. Gómez-Polo. 2016
51. Dryad Digital Repository.	Re-ordered in No.62	
Note: In the first row, fourteen references in gray shading are deleted; others are re-ordered with a new cited number, showing in orange. The references in blue are newly added, presented in the third row of the table.		
		53.A. Höskuldsson. 2011
		54.P. L. Harner. 2014
		57.H. R. N. Fonteles. 2020
		58.Z. Q. Zhang. 2019
		60.C. Alessia. 2017